# A genetically encoded probe for imaging nascent and mature HA-tagged proteins in vivo

Ning Zhao[1], Kouta Kamijo[2], Philip D. Fox [1], Haruka Oda[3], Tatsuya Morisaki [1], Yuko Sato[2,3], Hiroshi Kimura [2,3,4] & Timothy J. Stasevich[1,4]

To expand the toolbox of imaging in living cells, we have engineered a single-chain variable fragment binding the linear HA epitope with high affinity and specificity in vivo. The resulting probe, called the HA frankenbody, can light up in multiple colors HA-tagged nuclear, cytoplasmic, membrane, and mitochondrial proteins in diverse cell types. The HA frankenbody also enables state-of-the-art single-molecule experiments in living cells, which we demonstrate by tracking single HA-tagged histones in U2OS cells and single mRNA translation dynamics in both U2OS cells and neurons. Together with the SunTag, we also track two mRNA species simultaneously to demonstrate comparative single-molecule studies of translation can now be done with genetically encoded tools alone. Finally, we use the HA frankenbody to precisely quantify the expression of HA-tagged proteins in developing zebrafish embryos. The versatility of the HA frankenbody makes it a powerful tool for imaging protein dynamics in vivo.

[1] Department of Biochemistry and Molecular Biology, Colorado State University, Fort Collins, CO 80523, USA. [2] Graduate School of Life Science and Technology, Tokyo Institute of Technology, Yokohama 226-8503, Japan. [3] Cell Biology Center, Institute of Innovative Research, Tokyo Institute of Technology, Yokohama 226-8503, Japan. [4] World Research Hub Initiative, Institute of Innovative Research, Tokyo Institute of Technology, Yokohama 226-8503, Japan. Correspondence and requests for materials should be addressed to T.J.S. (email: Tim.Stasevich@colostate.edu)

L ive-cell imaging is critical for tracking the dynamics of cell signaling. The discovery and development of the green fluorescent protein (GFP), for example, has revolutionized the field of cell biology[1,2]. GFP can be genetically fused to a protein of interest (POI) to track its expression and localization in vivo. While powerful, GFP-tagging has limitations to image the full lifecycles of proteins. First, long fluorophore maturation times prevent co-translational imaging of GFP-tagged nascent peptide chains[3,4]. By the time the GFP tag folds, matures and lights up, translation is over. Similarly, slow GFP maturation times have made it difficult to image short-lived transcription factors critical for development and embryogenesis[5]. Again, before the GFP tag lights up, the transcription factor may have already been degraded. Second, GFP tags cannot discriminate post-translational modifications (PTM) to proteins[6], nor can they discriminate protein conformational changes[7]. Without the ability to directly image these important protein subpopulations, their functionality is difficult to quantify and assess. Third, GFP tags are large, permanently attached, and dim. It is therefore difficult to detect and/or amplify fluorescence signal. This limits the length of time a single tagged protein can be tracked in a living cell before the protein is photobleached or the cell is photodamaged.

To address these limitations of GFP, an alternative live-cell imaging methodology has emerged that uses antibody-based probes[8]. In this methodology, probes built from antibodies, such as antigen-binding fragments (Fabs)[9], single-chain variable fragments (scFvs)[10–12], and camelid nanobodies[13–16], are conjugated or genetically fused with mature fluorophores. When expressed or loaded into cells, the probes bind and light up epitopes within POIs as soon as the epitopes are accessible. With this methodology, it is possible to visualize and quantify the co-translational dynamics of nascent peptide chains[17–21], capture the dynamics of short-lived transcription factors[5], track single molecules for extended periods of time[22,23], and selectively track the spatiotemporal dynamics of PTMs[24] and protein conformational changes[7].

While there is potential for antibody-based probes in live-cell imaging, so far only a handful have been developed. Fab are straightforward to develop, since they can be digested from commercial antibodies and conjugated with dyes using kits[6,17,24,25]. Unfortunately, Fab have not been widely adopted because they are difficult to load into living systems. While adherent cells (e.g. U2OS) can be loaded in mass[26], sensitive cell types (e.g. neurons) have proven refractive to most loading procedures. Additionally, Fab are expensive, typically requiring milligrams of antibody to start with. Fab may also change considerably from batch to batch, which can lead to unwanted variability between experiments.

Given the drawbacks of Fab, genetically encoded probes are an attractive alternative. Since these probes can be integrated into plasmids, they can be distributed and cell lines and/or transgenic organisms can be generated that stably express the probes, all without batch-to-batch variability. A downside of genetically encoded antibody-based probes is they are not straightforward to develop. Both scFvs and nanobodies require a large initial investment, as either existing hybridomas or immunized animals are necessary to get the antibody sequences. Worse, even after sequences are determined, there is a good chance that antibody-based probes derived from the sequences will not fold and function properly in vivo. The problem is antibodies have evolved to be secreted from cells, so their folding and maturation is often disrupted when expressed within the reduced intracellular environment[27–29]. Thus, protein engineering, directed evolution, and mutagenesis are typically needed to generate an ideal antibody-based probe that functions in vivo.

A case in point is the SunTag scFv, the only genetically encoded antibody-based probe capable of binding a small epitope co-translationally in living cells. The SunTag scFv binds a 19 aa epitope (EELLSKNYHLENEVARLKK) that is repeated 24 times within a single SunTag. As multiple scFvs bind the SunTag co-translationally, fluorescence signal from tagged POIs can be amplified, enabling both single mRNA translation imaging and long-term single molecule tracking in vivo[18–21,30,31]. The SunTag technology was developed over many years, starting with the Plückthun lab in 1998. The original version was evolved through directed evolution and extensive protein engineering and later tested in 2014 to stain mitochondria in living cells[23,32,33]. The probe was further optimized via the addition of stabilizing sfGFP and GB1 domains to eliminate aggregation at higher expression levels and the original epitope was optimized to version 4 via directed mutagenesis.

The large amount of work required to develop the SunTag highlights the difficulty of generating scFv probes suitable for live-cell imaging. To confront this problem, we here develop an alternative strategy to bypass many of the difficulties. In our strategy, we begin with a diverse set of scFv scaffolds proven to fold and function in living cells. Onto these scaffolds, we loop graft all six complementarity determining regions (CDRs, or loops) from an epitope-specific antibody[34]. Depending on the compatibility of the scaffold and CDRs, this produces a hybrid scFv that retains the stability of the scaffold, while acquiring the specificity of the grafted CDRs.

To demonstrate our approach, we used it to generate two hybrid scFvs that bind the linear HA epitope (YPYDVPDYA)[35] in vivo. We used one of these hybrid scFvs, named HA frankenbody, to label in multiple colors a variety of proteins in diverse live-cell environments, including HA-tagged nuclear, cytoplasmic, membrane, and mitochondrial proteins. The HA frankenbody can also be used to track single 1 × HA-tagged proteins, to visualize HA-tagged protein translation, and to quantify HA-tagged protein expression in zebrafish embryos. The versatility of the HA frankenbody makes it a powerful tool for studying complex protein dynamics in living systems with high spatiotemporal resolution.

## Results

**Design strategy and initial screening of frankenbodies**. We engineered the HA-frankenbody from six CDRs of a published anti-HA scFv (parental full-length antibody: 12CA5)[36,37] (Fig. 1a). This wildtype anti-HA scFv (12CA5-scFv) does not fold properly in the reduced intracellular environment, and therefore displays little affinity for HA epitopes in living cells[36]. We hypothesized we could address this folding issue by grafting the CDRs onto more stable and similar scFv scaffolds (Fig. 1a). To test this, we selected five scFv scaffolds that have already been successfully used for live-cell imaging and that have a wide range of sequence identity compared to the 12CA5-scFv. In particular, the sequences of the heavy chain variable regions (VH) were 47–89% identical, while the sequences of the light chain variable regions (VL) were 50–67% identical (Table 1). The five scaffolds we chose included (1) an scFv-binding histone H4 mono-methylated at Lysine 20 (H4K20me; 15F11)[38]; (2) an H3K9ac-specific scFv (13C7)[39]; (3) an H4K20me2-specific scFv (2E2, unpublished); (4) a SunTag-specific scFv[23]; and (5) a bone Gla protein (BGP)-specific scFv (KTM219)[36]. Among these, 15F11 and 2E2 have the greatest sequence identity compared to the 12CA5-scFv.

We grafted the 12CA5-scFv CDRs onto our five scFv scaffolds. We refer to the resulting five chimeric scFvs as $\chi_{\text{scaffold}}^{\text{HA}}$. For example, $\chi_{\text{15F11}}^{\text{HA}}$ specifies the chimeric scFv that was generated by

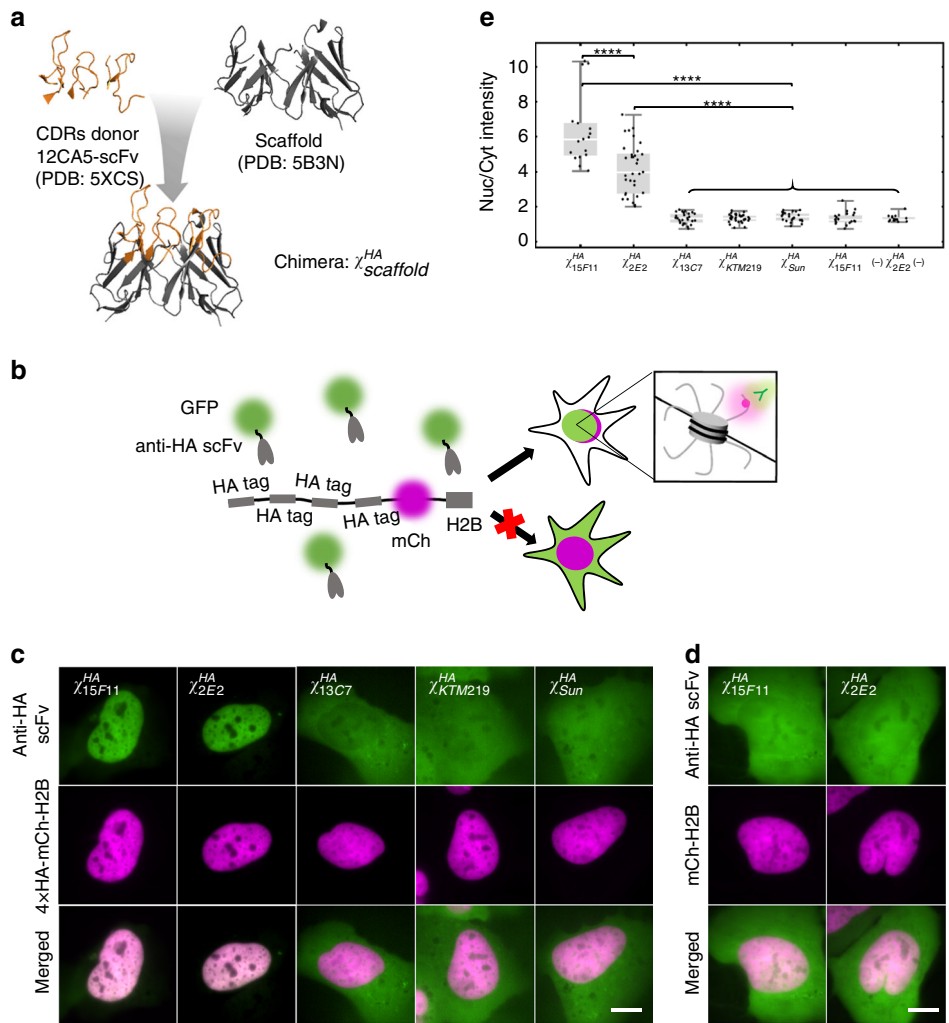

**Fig. 1** Design strategy and initial screening of frankenbodies. **a** A cartoon schematic showing how to design a chimeric anti-HA scFv using 12CA5-scFv CDRs and stable scFv scaffolds. **b** A cartoon showing how to screen the five chimeric anti-HA scFvs in living U2OS cells. **c** Initial screening results showing the respective localization of the five chimeric anti-HA scFvs in living U2OS cells co-expressing HA-tagged histone H2B (chimeric anti-HA scFv, green; 4 × HA-mCh-H2B, magenta). From left to right, $n = 17, 33, 27, 31, 25$ cells. **d** Control results showing the respective localization of $\chi^{HA}_{15F11}$ and $\chi^{HA}_{2E2}$ in living cells lacking HA-tagged histone H2B (chimeric anti-HA scFv, green; mCh-H2B, magenta). From left to right, $n = 21, 15$ cells. **e** Nuclear to cytoplasmic fluorescent intensity ratio (Nuc/Cyt) plot of each chimeric anti-HA scFv for all cells imaged as in **c** and **d**. Student's $t$-test. ****$p < 0.0001$. All images are representative cell images from one independent experiment. Scale bars: 10 μm. Source data are provided as a Source Data file. For the box and whisker plots, median is shown by a white line, the box indicates 25–75% range, and whiskers indicate 5–95% range

### Table 1 Sequence identity analysis

| Scaffolds | CDRs donor 12CA5-scFv | |
| --- | --- | --- |
| | Sequence identity (%) | |
| | VH | VL |
| 15F11 | 85 | 67 |
| 2E2 | 89 | 65 |
| 13C7 | 47 | 57 |
| KTM219 | 47 | 65 |
| Sun | 63 | 50 |

loop grafting the 12CA5-scFv CDRs onto the 15F11 scaffold. To screen our chimeras, we fused each with the monomeric enhanced GFP (mEGFP) and co-transfected each of the resulting plasmids into U2OS cells, together with a plasmid encoding 4 × HA-tagged red fluorescent protein mCherry fused to histone H2B (4 × HA-mCh-H2B). If a chimeric scFv binds to the HA epitope

in living cells, it should co-localize with the HA-tagged H2B in the nucleus, as shown in Fig. 1b. Live-cell imaging revealed $\chi^{HA}_{15F11}$ and $\chi^{HA}_{2E2}$ (sequences in Supplementary Fig. 1) were superior, displaying little to no misfolding and/or aggregation, strong expression, and co-localization with H2B in the nucleus. In contrast, the other three scFvs did not show any co-localization (Fig. 1c, e). Moreover, in control cells lacking HA tags, both $\chi^{HA}_{15F11}$ and $\chi^{HA}_{2E2}$ displayed uniform expression (Fig. 1d, e), indicative of free diffusion without non-specific binding. According to our screen, both $\chi^{HA}_{15F11}$ and $\chi^{HA}_{2E2}$ function well, although $\chi^{HA}_{15F11}$ labels HA tags slightly better than $\chi^{HA}_{2E2}$ (Fig. 1e). We therefore chose the $\chi^{HA}_{15F11}$ variant for additional characterization, which we herein refer to as the HA frankenbody due to its construction via grafting.

**Multicolor labeling of HA-tagged proteins in vivo.** We tested the HA frankenbody in a variety of different settings. First, since the initial screening had been done with a 4 × HA tag, we wanted

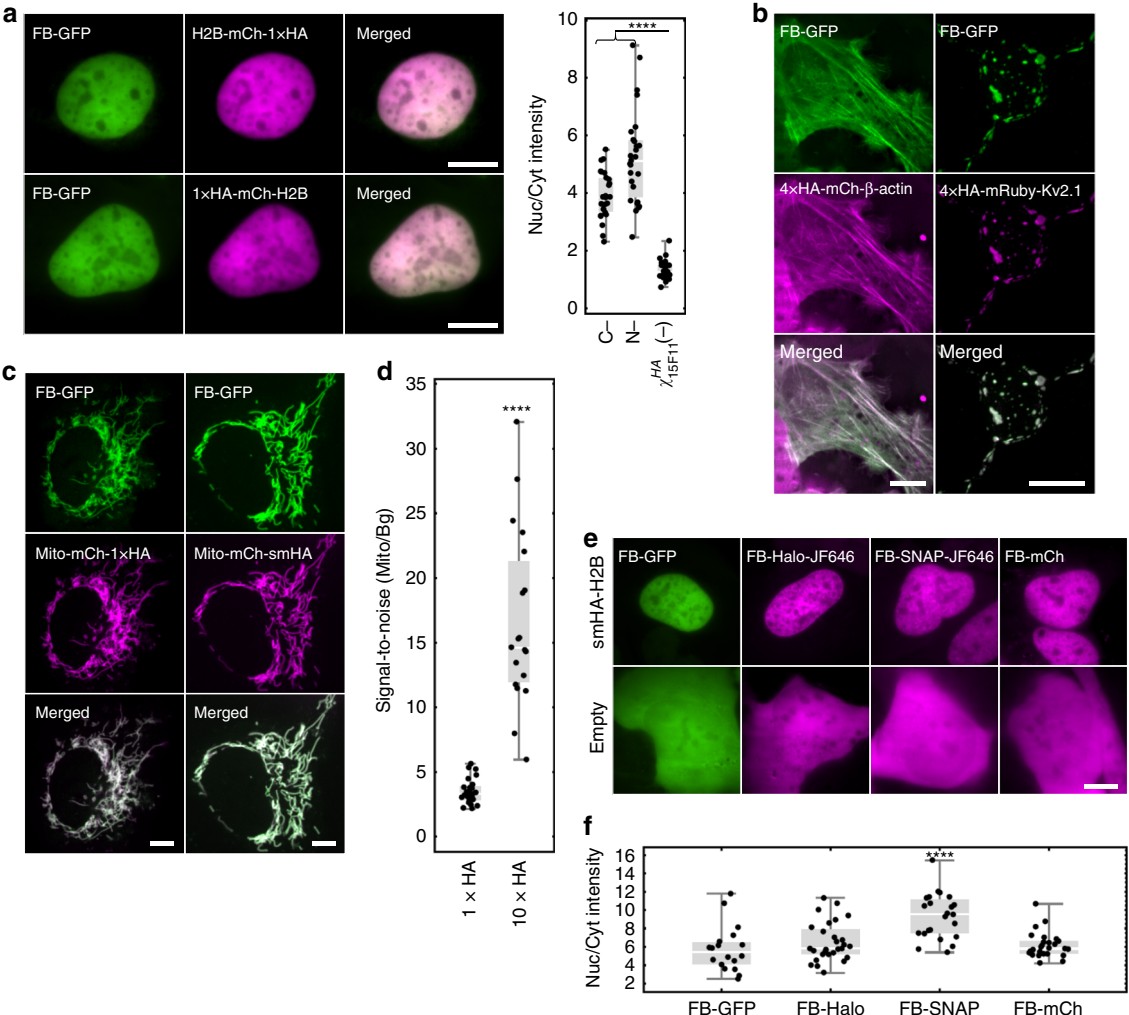

**Fig. 2** Multicolor labeling of HA-tagged proteins in vivo. **a** Frankenbody (FB-GFP; green) labels a 1 × HA-tagged nuclear protein, histone H2B (magenta), at the N-terminus or C-terminus in living U2OS cells. Left top: 1 × HA at C-terminus (H2B-mCh-1 × HA, $n = 27$); Left bottom: 1 × HA at N-terminus (1 × HA-mCh-H2B, $n = 27$). Right: Nuclear to cytoplasmic fluorescent intensity ratio (Nuc/Cyt) plot of all imaged cells. **b** FB-GFP (green) labels HA-tagged cytoplasmic protein β-actin (4 × HA-mCh-β-actin, magenta, $n = 18$) in living U2OS cells and membrane protein Kv2.1 (4 × HA-mRuby-Kv2.1, magenta, $n = 13$) in living neurons. See also Supplementary Movie 1. **c** FB-GFP (green) labels 1 × HA or 10 × HA spaghetti monster (smHA) tagged mitochondrial protein mitoNEET (Mito, magenta) in living U2OS cells. Left: Mito-mCh-1 × HA ($n = 24$); right: Mito-mCh-smHA ($n = 19$). **d** Mitochondria to background fluorescent intensity ratio (Mito/Bg) plot of all cells imaged as in **c**. Mean of Mito/Bg = 3.5 ± 0.2 (mean ± SEM) for 1 × HA and 16.6 ± 1.5 (mean ± SEM) for smHA. This result shows the Mito/Bg ratio is 4.7 ± 0.5 (SEM) times higher for smHA tagged Mito than 1 × HA tagged Mito. **e** FB fused to multiple fluorescent fusion proteins specifically labels HA-tagged nuclear protein H2B (smHA-H2B). Top row: GFP, HaloTag-JF646, SNAP-tag-JF646 and mCherry; from left to right, $n = 18, 28, 23, 26$ cells. In all cases, cells lacking the HA-tag display relatively even FB fluorescence. Bottom row: from left to right, $n = 21, 19, 11, 14$ cells. See also Supplementary Fig. 2. **f** Nuclear to cytoplasmic fluorescent intensity ratio (Nuc/Cyt) plot of all cells imaged as in the top row of **e**. All images are representative cell images in one independent experiment. Student's $t$-test. ****$p < 0.0001$. Scale bars, 10 μm. Source data are provided as a Source Data file. For the box and whisker plots, median is shown by a white line, the box indicates 25–75% range, and whiskers indicate 5–95% range

to see if the HA frankenbody could also bind a 1 × HA tag. To test this, we constructed two plasmids: 1 × HA fused to the C-terminus of H2B-mCherry (H2B-mCh-1 × HA) and 1 × HA fused to the N-terminus of mCherry-H2B (1 × HA-mCh-H2B). In both cases, the HA frankenbody displayed strong nuclear localization (Fig. 2a). Beyond nuclear proteins, we also wanted to test if the HA frankenbody can work well in the cell cytoplasm, another reducing environment that can interfere with disulfide bond formation[33]. We tested this by creating a target plasmid encoding the cytoplasmic protein β-actin fused with a 4 × HA-tag and mCherry (4 × HA-mCh-β-actin). When this plasmid was expressed in cells, co-expressed frankenbodies again took on the localization pattern of their targets, now colocalizing with 4 × HA-mCh-β-actin along filamentous actin fibers (Fig. 2b, left). We

therefore conclude that both nuclear and cytoplasmic HA-tagged proteins can be labeled by the HA frankenbody in living cells.

To test if frankenbodies could also work in more sensitive cells, we co-transfected living neurons with the HA frankenbody (FB-GFP) and a 4 × HA-tagged transmembrane protein Kv2.1 (4 × HA-mRuby-Kv2.1). In neurons, Kv2.1 has been demonstrated to localize to the plasma membrane, where it forms cell-surface clusters[40]. In cells expressing FB-GFP and 4 × HA-mRuby-Kv2.1, the HA frankenbody again took on the localization pattern of its target (Fig. 2b right and Supplementary Movie 1). In addition, the pattern could be seen for over a week after transient transfection (Supplementary Fig. 2 left). This demonstrates the HA franken-body can bind HA-tagged membrane proteins, as well as cytoplasmic and nuclear proteins. This also demonstrates

frankenbodies have long half-lives and their continual expression does not detrimentally impact sensitive cells.

Because the HA epitope is so small, it can be repeated within tags to increase signal-to-noise. The spaghetti monster HA tag (smHA), for example, contains 10 HA epitopes and has been used to amplify fluorescence signal from tagged proteins[22]. To test how well the frankenbody labels smHA, we co-transfected cells with a plasmid encoding either a $1 \times$ HA or smHA fused to an mCherry-tagged mitochondrial protein, mitoNEET[41] (Mito). As before, the HA frankenbody colocalized with target proteins (Fig. 2c, d). Moreover, quantification revealed the mitochondrial signal-to-noise (Mito/Bg) was on average $4.7 \pm 0.5$ (mean $\pm$ SEM) times higher for smHA-tagged Mito compared to $1 \times$ HA-tagged Mito. Thus, the HA frankenbody can be used to amplify fluorescence in live-cell imaging.

Finally, to ensure the HA frankenbody is as broadly applicable as possible, we tested if it could tolerate different fusion partners for multicolor imaging. We fused HA frankenbody to mCherry, HaloTag[42], and SNAP-tag[43]. All three frankenbody constructs colocalized with HA-tagged H2B in the nucleus of living U2OS cells, similar or even better than the original GFP-tagged frankenbody (Figs. 2e upper, f). Furthermore, when different colored frankenbodies were co-expressed in cells, one color did not dominate over the others (Supplementary Fig. 2, right). Finally, all three constructs displayed diffuse localization patterns in cells lacking the HA-tag (Fig. 2e lower). These data indicate the HA frankenbody can label HA-tagged proteins in a rainbow of colors in living cells.

**Using purified recombinant frankenbody in vitro**. We wondered if the HA frankenbody has the potential to replace costly anti-HA antibodies in assays such as immunostaining and Western blots. To test this, we purified recombinant frankenbody from *E. coli* and used it to immunostain fixed cells expressing HA-tagged H2B or β-actin. The purified HA frankenbody stained both the HA-tagged nuclear and cytoplasmic proteins with almost no background (Fig. 3a, b).

We next tested the suitability of the HA frankenbody for Western blotting. For this, we harvested U2OS cells expressing HA-tagged H2B or β-actin. In contrast to the parental 12CA5 anti-HA antibody, which we had to stain with a secondary antibody, we could detect the frankenbody in blots using the GFP signal alone. Similar bands were seen in both cases (Fig. 3c and Supplementary Fig. 3). Although several of the bands were dimmer than those using the antibody, we attributed the difference to signal amplification from the secondary antibody. Together, our Western blot and immunostaining results demonstrate the HA frankenbody can serve as a cost-effective replacement for the anti-HA antibody in widely used in vitro applications.

**HA frankenbody binds the HA epitope for minutes in cells**. An ideal imaging probe binds its target with high affinity to maximize the fraction of target epitopes bound and thereby increase signal-to-noise. With this in mind, we set out to measure the length of time the HA frankenbody remains bound to the HA epitope in living cells. For this, we performed fluorescence recovery after photobleaching (FRAP) experiments in cells co-expressing GFP-tagged HA frankenbody and $4 \times$ HA-mCh-H2B. As H2B is bound to chromatin for hours at a time[44], any recovery on the minutes timescale can be attributed to the turnover of frankenbody alone. Consistent with this, FRAP in the $4 \times$ HA-mCh-H2B channel displayed little or no recovery on the timescale of our experiment. In contrast, FRAP in the FB-GFP channel slowly recovered (Fig. 4a). Quantitative analyses of FRAP curves revealed the half

recovery time to be 2–3 min (Fig. 4b, d). As a control, we repeated FRAP experiments in cells transfected with just the HA frankenbody (i.e. lacking HA epitopes). Here, the FRAP lasted just seconds, consistent with little to no non-specific binding of the HA frankenbody (Fig. 4c, d). We therefore conclude that the majority of frankenbodies bind target HA epitopes for minutes at a time in living cells. This is consistent with the high binding affinity ($K_D = 14.7 \pm 7.4$ nM; an order of magnitude higher than the scaffold 15F11 scFv[38]) measured in vitro using surface plasmon resonance (Supplementary Fig. 4).

**Single molecule tracking of 1×HA-tagged proteins in cells**. The long binding time of the HA frankenbody to HA epitopes in living cells suggested the frankenbody can be used to track single HA-tagged proteins. To demonstrate it, we tracked $1 \times$ HA-mCh-H2B proteins in cells with FB-Halo-TMR. To increase the number of tracks, TMR was pre-treated with a reducing agent that causes it to photoactivate via 405 nm stimulation[45]. This enabled denser single molecule tracking for super-resolution imaging in living cells[15].

In each cell, we localized up to $\sim 10^6$ FB-Halo-TMR, generating $10^3–10^4$ tracks per cell (Supplementary Movie 2 and Supplementary Fig. 5a). Consistent with our live-cell images in Fig. 2a, nearly all FB-Halo tracks were within the nucleus (Supplementary Fig. 5a). We filtered tracks based on mobility to focus on just the chromatin-bound FB-Halo[46]. This left $10^3–10^4$ tracks, which was orders of magnitude greater than in control cells lacking $1 \times$ HA-mCh-H2B (Fig. 5b and Supplementary Fig. 5b). From the filtered tracks, we generated a map of the mobility of histones across the nucleus (Fig. 5a). This map revealed histones near the nuclear periphery have reduced mobility, consistent with recent single-molecule experiments using photoactivatable H2B-PAmCh[47]. Moreover, the average mean squared displacement of our filtered $1 \times$ HA-mCh-H2B tracks displayed similar diffusivity as H2B-PAmCh[47] (Fig. 5c). We therefore conclude the HA frankenbody can be used to track $1 \times$ HA-tagged proteins, provided their mobility and/or localization is distinct from unbound frankenbody.

**Tracking single mRNA translation in living U2OS cells**. A major advantage of the HA frankenbody over other intrabodies is the small size and linearity of its epitope, just 9 aa in length. This means the epitope is quickly translated by the ribosome and becomes available for binding almost immediately. The HA frankenbody therefore has the potential to bind HA-tagged nascent peptides co-translationally[17]. By simply repeating the HA epitope multiple times within a tag, fluorescence can furthermore be amplified for enhanced tracking[22].

To test the potential of HA frankenbody for imaging translation dynamics, we co-transfected the FB-GFP into U2OS cells together with our translation reporter smHA-KDM5B-MS2. The reporter encodes a smHA tag N-terminally fused to the nuclear protein KDM5B. In addition, the reporter contains a $24 \times$ MS2 stem loop repeat in the 3′ UTR to label and track single mRNA (Fig. 6a). A few hours after transfection, single mRNA (labeled by HaloTag fused MS2 coat protein, MCP-HaloTag, and the JF646 HaloTag ligand[48]) could be seen diffusing throughout the cell cytoplasm. The HA frankenbody co-moved with many mRNA (Fig. 6b, Supplementary Movie 3), indicative of active translation[17]. To prove these were translation sites, we added the translational inhibitor puromycin. This caused the frankenbody to disperse from all single mRNA sites within seconds (Fig. 6c, Supplementary Movie 4), confirming the HA frankenbody can bind nascent peptides co-translationally.

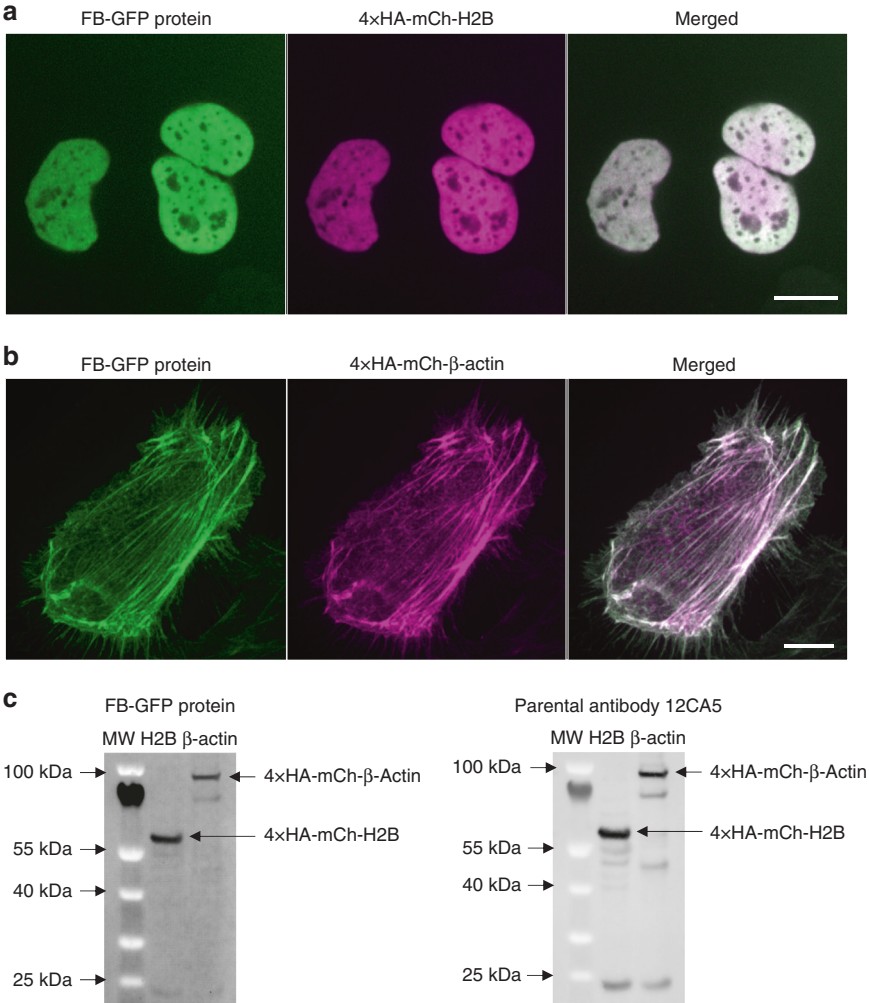

**Fig. 3** Using purified recombinant frankenbody in vitro. Immunostaining in fixed U2OS cells with purified frankenbody (FB-GFP; green) of an HA-tagged (**a**) nuclear protein, histone H2B (4 × HA-mCh-H2B; magenta; a representative cell image of $n = 27$ cells in one independent experiment) and (**b**) cytoplasmic protein, β-actin (4 × HA-mCh-β-actin; magenta; a representative cell image of $n = 10$ cells in one independent experiment). **c** Western blot of HA-tagged H2B and β-actin. Left: purified FB-GFP (1:2000 dilution, no secondary antibody) detected directly using GFP fluorescence; Right: parental anti-HA antibody 12CA5 (1:2000 dilution) detected with secondary anti-mouse antibody/Alexa488 (1:5000 dilution). Scale bars, 10 μm

To check if the frankenbody can light up translation sites in multiple colors, we repeated experiments, but now using other frankenbody constructs. For this, we co-transfected cells with mCherry, HaloTag, or SNAP-tag frankenbody (FB-mCh, FB-Halo, or FB-SNAP), together with the translation reporter (Fig. 6d). For mCherry and HaloTag, we could detect translation sites that responded to puromycin (Fig. 6e,f, Supplementary Movies 5 and 6). With SNAP-tag, results were less clear. Although we could see some bright puncta in cells, not all puncta responded to puromycin (suggesting there may be non-specific aggregation induced by the SNAP-tag). Nevertheless, the data demonstrate we can image translation in three colors spanning the imaging spectrum.

**Multiplexed imaging of single mRNA translation dynamics.** With the ability to image translation in more than one color, we combined the HA frankenbody with the SunTag to simultaneously quantify the translation kinetics of two mRNA species in single living cells. Previously, the SunTag scFv has been fused to GFP (Sun-GFP) to monitor translation[18–21]. We therefore coupled this probe[23] (after removing its HA epitope) with our complementary mCherry-tagged frankenbody probe (FB-mCh) (Fig. 7a). Co-transfecting these into living U2OS cells with plasmids encoding SunTag-kif18b and smHA-KDM5B-MS2, we observed translation sites labeled by either Sun-GFP alone or FB-mCh alone (Fig. 7b, Supplementary Movie 7). After co-tracking hundreds of translation sites, we quantified their mobilities. This revealed both types of translation sites have similar diffusion coefficients (over short timescales): $0.016 \pm 0.004$ μm²/s (95% CI) for FB-mCh and $0.019 \pm 0.006$ μm²/s (95% CI) for Sun-GFP. The similarity of their movement despite their distinct sequences shows that different mRNA can be translated in similar micro-environments.

Since both translation sites were labeled in different colors, we next wondered if they ever co-localized. The observation of co-localized translation sites would provide further evidence for multi-RNA translation factories, which our group[17] and another[21] have recently observed. Although we looked, we were unable to detect co-localized FB-mCh and Sun-GFP translation sites. Since the smHA-KDM5B-MS2 mRNA has different 3′ and 5′ UTRs, as well as a different ORF than SunTag-kif18b mRNA, these data suggest that the composition of factories may be dictated in part through mRNA sequence elements.

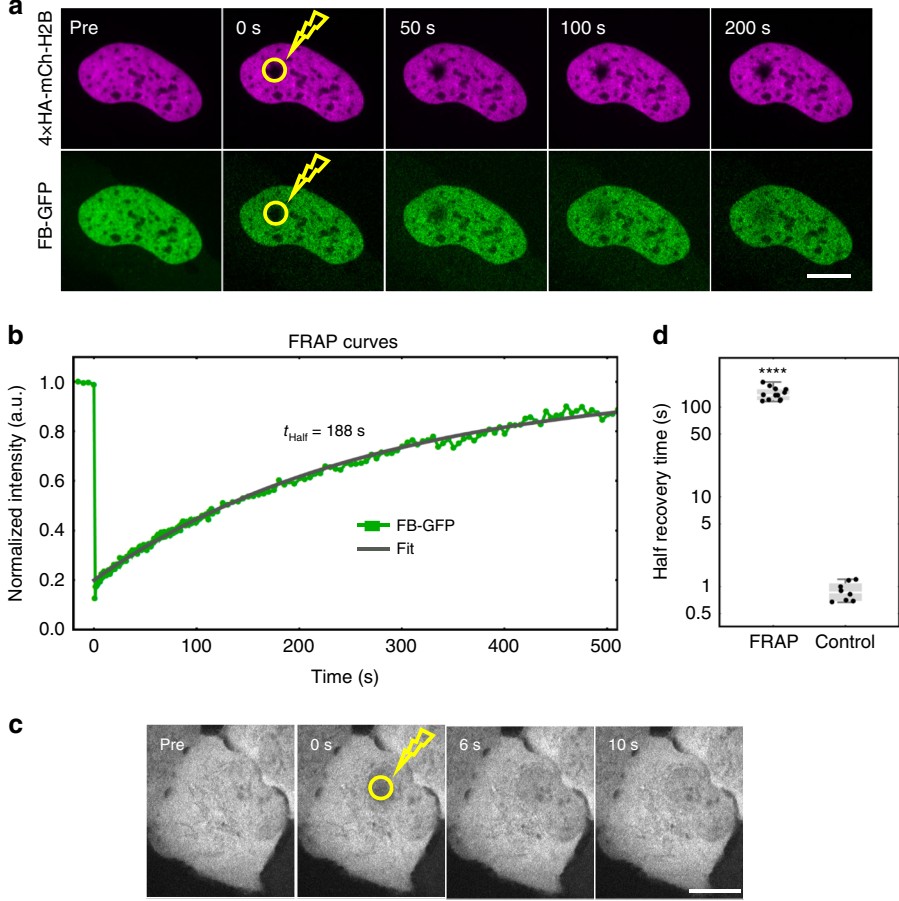

**Fig. 4** HA frankenbody binds the HA epitope for minutes in cells. **a** A representative FRAP experiment (yellow circle indicates bleach spot) showing fluorescence recovery in cells expressing frankenbody (FB-GFP; green) and target 4 × HA-mCh-H2B (magenta). **b** Quantification of FRAP data in a representative cell, along with a fitted curve. **c** A representative FRAP experiment (yellow circle indicates bleach spot) in cells expressing FB-GFP only (i.e. cells lacking HA-tags) is complete in less than 10 s. **d** Half recovery time plot of FRAP experiments (FB-GFP and 4 × HA-mCh-H2B, $n = 12$ cells in 2 independent experiments, as in **b**) and controls (FB-GFP only, $n = 8$ cells in one independent experiment, as in **c**). Fits from 12 cells reveal the FRAP mean half recovery time, $t_{half}$, is 141 ± 7 s (cell-to-cell SEM). Student's $t$-test. ****$p < 0.0001$. Scale bars, 10 μm. Source data are provided as a Source Data file. For the box and whisker plots, median is shown by a white line, the box indicates 25–75% range, and whiskers indicate 5–95% range

**Monitoring local translation in living neurons**. Local translation is implicated in neuronal plasticity, memory formation, and disease[49]. The ability to image local translation at the single molecule level in living neurons would therefore be a valuable research tool to better understand these processes. To facilitate this, we tested if the HA frankenbody could be used to monitor single mRNA translation in living primary neurons. When we co-transfected these cells with our KDM5B reporter[17] and the FB-GFP (Fig. 8a), we were able to see distinct bright spots that diffused throughout the cell cytoplasm (Fig. 8b, Supplementary Movie 8), reminiscent of the translation sites we had observed in U2OS cells. Again, we confirmed these were translation sites by adding puromycin (Fig. 8c, Supplementary Movie 9). Just seconds after puromycin was added, the bright spots disappeared, just as they had in U2OS cells.

Unlike the mobility of mRNA we observed in U2OS cells, mRNA within neuronal dendrites displayed obvious directed motion events. For example, we regularly saw mRNA zip along linear paths within dendrites, achieving rapid translocations with rapid retrograde and anterograde transport over large distances up to 8 μm (Fig. 8d,e, and Supplementary Fig. 6, Supplementary Movie 8). The frankenbody signal within these fast-moving sites suggest translation is still active, despite the motored movement. These data therefore provide support for a model in which

translation is not repressed during trafficking[20,21]. Also, since the KDM5B translation reporter is the same as the one we used in U2OS cells (Figs. 6 and 7), these data suggest the cell type and/or local environment plays a significant role in dictating translation site mobility.

**Monitoring HA-tagged proteins in zebrafish embryos**. Arguably the most demanding types of imaging applications are in whole living animals. To verify that the HA frankenbody can also be applied in this way, we used it to monitor development in zebrafish embryos. The environment within embryos is complex and contains many potential non-specific binding targets. To express HA frankenbody in this complex environment, we microinjected mRNA encoding the FB-GFP and HA-mCh-H2B into the yolk of one-cell stage zebrafish eggs. With this setup, HA frankenbody is translated immediately without having to wait for the onset of transcription after the maternal-zygotic transition[50]. Following the initial injection, we co-loaded a positive control Fab (Cy5 conjugated) which specifically binds and lights up endogenous histone acetylation (H3K9ac) in the cell nucleus[25] (Supplementary Fig. 7a).

We began imaging embryo development with the HA frankenbody around the four-cell or eight-cell stage. At all time points we could see colocalization of the frankenbody with the

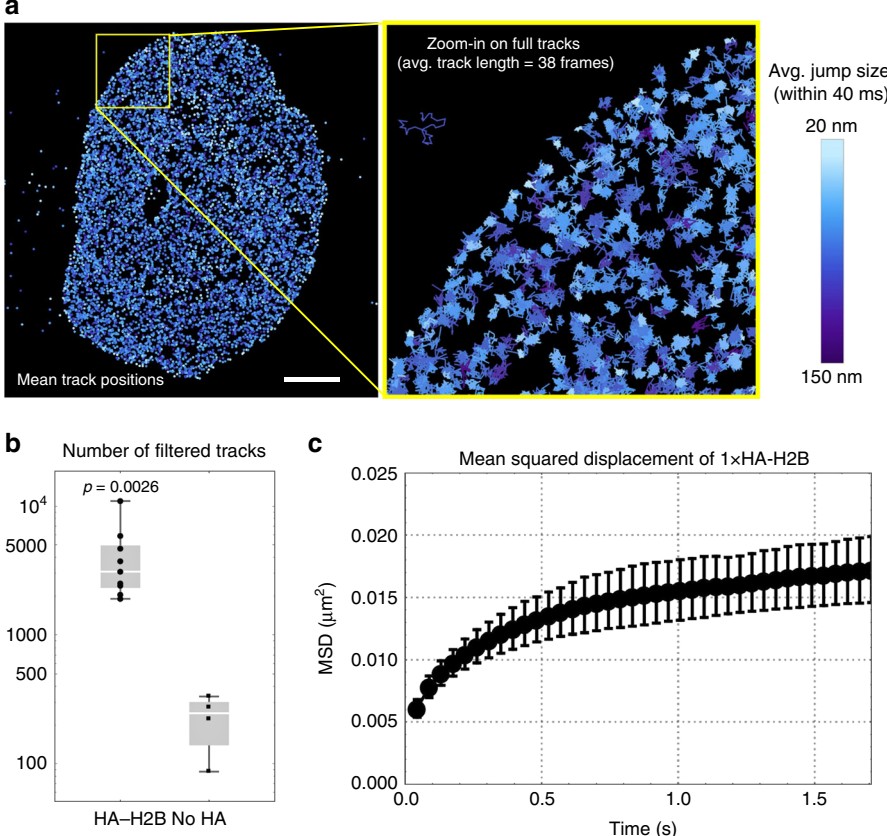

**Fig. 5** Single molecule tracking of 1 × HA-tagged proteins in cells. **a** The mean positions of tracks of single frankenbody (FB) bound to 1 × HA-H2B provides a mobility map of H2B across the cell nucleus (10,949 tracks were generated from 977,516 total FB localizations). Tracks are color coded according to their average frame-to-frame jump size. The lighter colored tracks with relatively small jump sizes are enriched along the edge of the cell nucleus, where heterochromatin is typically enriched. To ensure tracks represent FB bound to HA-H2B, tracks were filtered such that their length had to be at least 16 consecutive frames and jumps between frames had to all be <220 nm. Full tracks within the yellow box are displayed in the zoom-in on the right. See also Supplementary Movie 2. **b** In cells expressing HA-H2B, the number of filtered FB tracks were between one and two orders of magnitude greater than in control cells lacking HA-H2B, demonstrating false-positive tracks are rare (see Supplementary Fig. 5). **c** The average mean-squared displacement of tracks provides a good estimate for average HA-H2B mobility. All tracks are from $n = 9$ cells in three independent experiments. Student's $t$-test. Error bar, cell-to-cell SD of the average MSD. Scale bar, 5 μm. Source data are provided as a Source Data file. For the box and whisker plots, median is shown by a white line, the box indicates 25–75% range, and whiskers indicate 5–95% range

4 × HA-mCh-H2B target (Fig. 9a, Supplementary Movie 10), although at earlier time points the concentrations of both were lower and therefore marked the nuclei only dimly compared to the positive control Fab. Nevertheless, we could detect all three signals in the nuclei of single mother and daughter cells throughout the entire 80-min time course (Fig. 9b upper). Moreover, the signal from the frankenbody in the nucleus correlated with the target 4 × HA-mCh-H2B signal, increasing from a nuclear-to-cytoplasmic ratio of one to nearly 2.5 (Fig. 9b lower). This single-cell trend was also observed in the population of cells (Fig. 9c). As a negative control, we repeated experiments in zebrafish embryos lacking target 4 × HA-mCh-H2B. In this case, the frankenbody was evenly distributed throughout the embryo (Supplementary Fig. 7b, c, Supplementary Movie 11), displaying a nuclear-to-cytoplasmic ratio close to one. To see how signal varies with the number of HA epitopes, we repeated experiments with N × HA-mCh-H2B (N = 1, 4, 10; Supplementary Fig. 8). In all cases, we could track cell nuclei with HA frankenbody, and signal improved as more epitopes were added to the tag. In fact, for N = 10, the HA frankenbody signal was significantly brighter than the target mCherry-H2B signal (Supplementary Fig. 8). This confirms the HA frankenbody binds the HA epitope selectively and tightly in vivo, so it can be

used to monitor the concentration of target HA-tagged proteins in living organisms.

## Discussion

While scFvs have great potential for live-cell imaging, so far there are just a few scFvs that fold and function in living cells. Here we employed CDR loop grafting to generate with a 40% success rate stable and functional scFvs that bind the linear HA epitope tag in vivo, one of which we named the HA frankenbody and further characterized. The HA frankenbody is capable of labeling HA-tagged nuclear, cytoplasmic, membrane, and mitochondrial proteins in multiple colors and in a diverse range of cellular environments.

A major advantage of the HA frankenbody is it can be used to image single mRNA translation dynamics in living cells. This is because the target HA epitope (YPYDVPDYA, 9 aa) is small and linear. It therefore emerges quickly from the ribosome, so it can be co-translationally labeled by frankenbody almost immediately. It is also short enough to repeat many times in a single tag for signal amplification, as in the smHA tag[22]. In principle, epitopes could even be made conditionally accessible within a protein to monitor conformational changes. In contrast, almost all other antibody-based probes bind to 3D

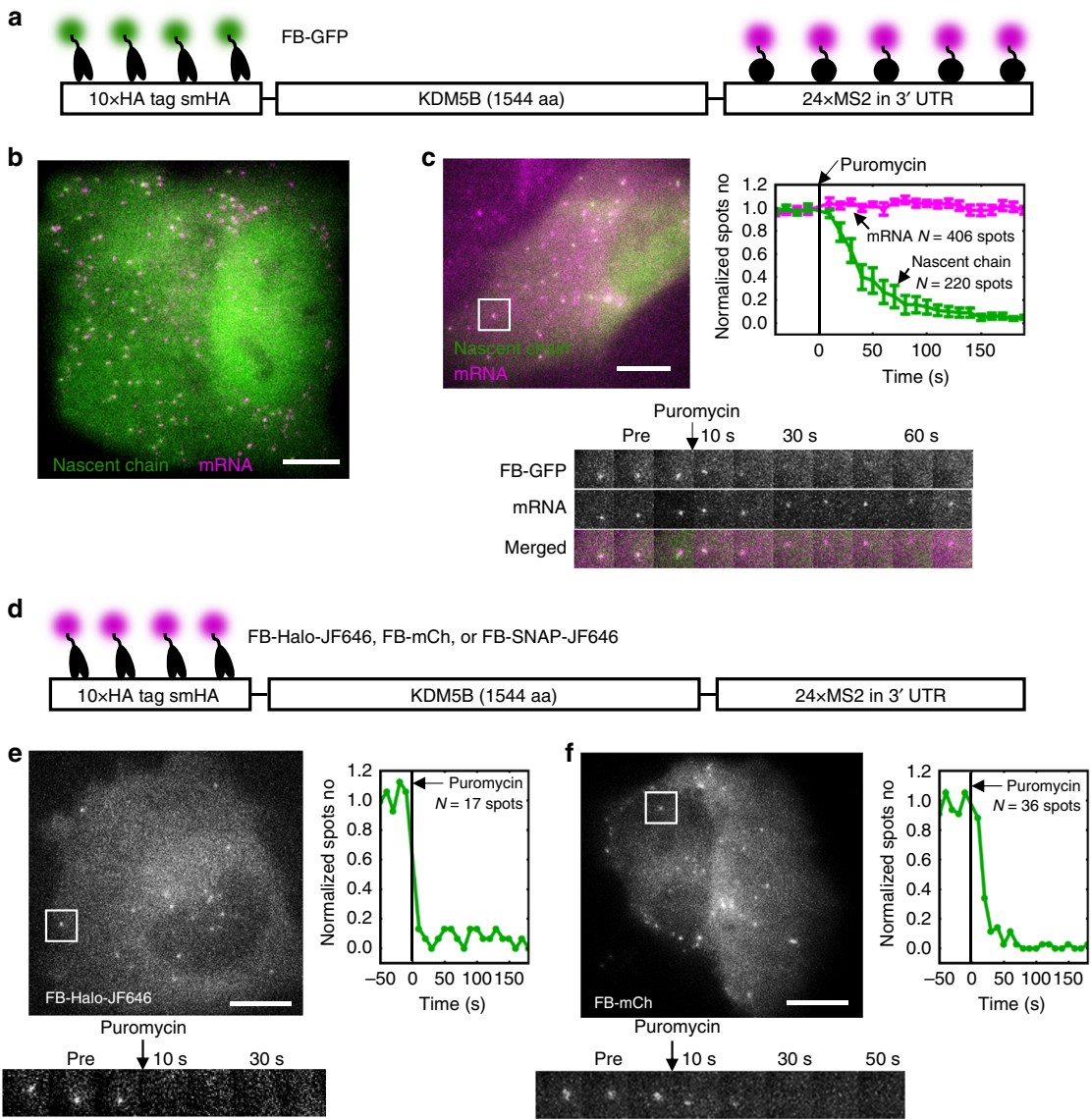

**Fig. 6** Tracking single mRNA translation in living U2OS cells. **a** A diagram depicting frankenbody (FB-GFP; green) and MCP-HaloTag-JF646 (magenta) labeling HA epitopes and mRNA stem loops, respectively, in a KDM5B translation reporter. **b** A representative cell (10 cells in three independent experiments) showing colocalization of FB-GFP (green) with KDM5B mRNA (magenta). See also Supplementary Movie 3. **c** A representative cell (upper-left, 9 cells in three independent experiments) showing the disappearance of nascent chain spots labeled by FB-GFP within seconds of adding the translational inhibitor puromycin. See also Supplementary Movie 4. Upper-right: The mean number of nascent chain spots normalized to pre-puromycin levels decreases while mRNA levels remain constant (9 cells from three independent experiments). Error bars, cell-to-cell SEM. Lower: a sample single mRNA montage. **d** A diagram depicting FB-Halo-JF646, FB-mCh, or FB-SNAP-JF646 labeling HA epitopes in a KDM5B translation reporter. **e**, **f** Representative cells (three cells in two independent experiments for both FB-Halo and FB-mCh), single mRNA montages, and quantification, as in **c**, showing the loss of nascent chain spots labeled by **e** FB-Halo-JF646 or **f** FB-mCh upon puromycin treatment. See also Supplementary Movies 5 and 6. Scale bars, 10 μm. Source data are provided as a Source Data file

epitopes that span a large length of linear sequence space. In general, 3D epitopes take a relatively long time to be translated and fold before they become accessible for probe binding. Furthermore, they are too big to repeat more than a few times, so fluorescence is difficult to amplify for single molecule tracking[22,23].

Here we used the HA frankenbody to image single mRNA translation in both living U2OS cells and primary neurons. Unlike Fab, which cause neurons to peel during the loading procedure, HA frankenbody can be expressed in neurons without issue via transfection. We exploited this to demonstrate the mobility of translating mRNA is cell-type dependent. While our KDM5B translation reporter mRNA displayed largely non-

directional, diffusive movement in U2OS cells, in neurons they were often motored. Neurons can be notoriously long, so motored mRNA movement provides a solution to the unique challenge of local protein production in distal neuronal dendrites and axons[51,52]. An open question is if translation is repressed during transport. On the one hand, certain mRNA are known to be actively repressed during trafficking[3,53], perhaps to conserve energy. On the other hand, the Singer and Bertrand labs have recently shown that mRNA can be actively translated during transport[20,21]. Here we see a similar phenomenon, with our KDM5B translation reporter being rapidly motored over distances up to 8 μm in dendrites while retaining a strong translation signal.

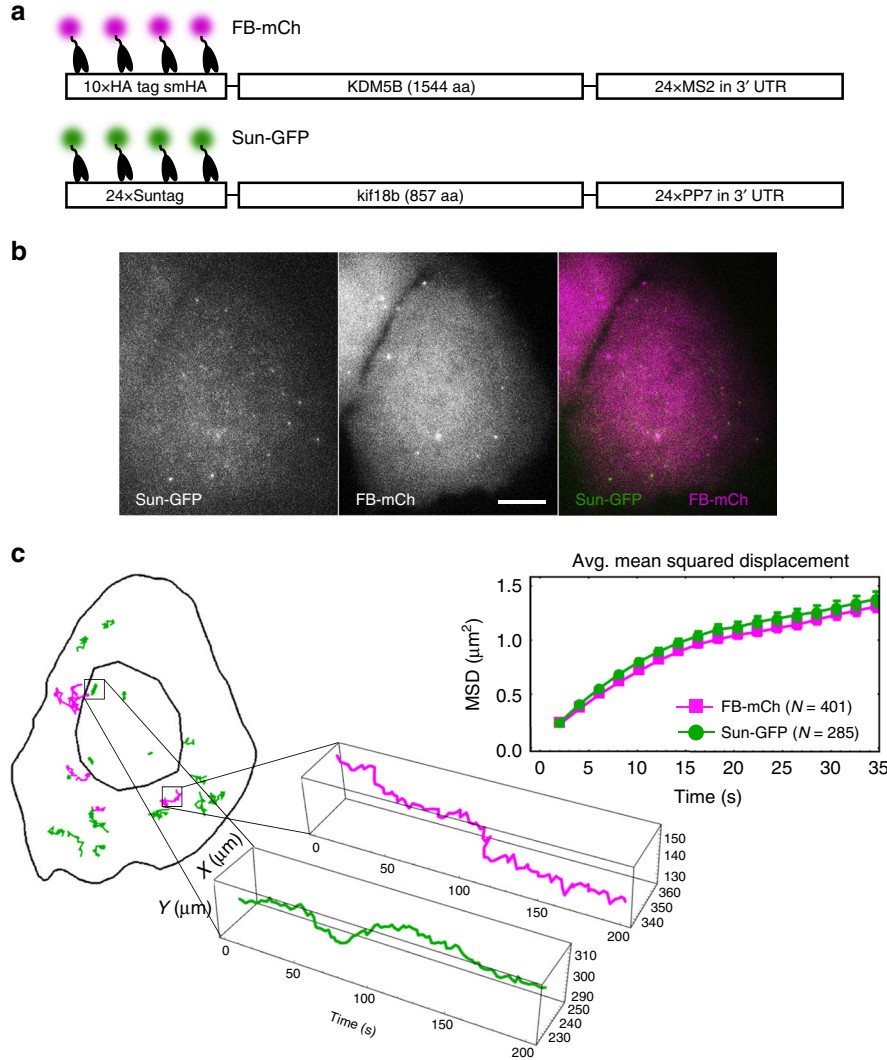

**Fig. 7** Multiplexed imaging of single mRNA translation dynamics. **a** A diagram depicting frankenbody (FB-mCh) and Sun-GFP-labeling epitopes in the KDM5B and kif18b translation reporter constructs, respectively. **b** A representative living U2OS cell (n = 8 cells from three independent experiments) showing nascent chain translation spots labeled by FB-mCh (magenta) or Sun-GFP (green). **c** The mask and tracks of the cell in **b**, and dynamics of a representative translation spot for each probe (Sun-GFP, green; FB-mCh, magenta). Right, the average mean squared displacement of FB-mCh (magenta) and Sun-GFP (green) translation sites (upper-right) from N = 401 tracks for FB-mCh, N = 285 tracks for Sun-GFP in eight cells in three independent experiments. Error bars, RNA-to-RNA SEM. Fits to the first five points of the MSD curves show the diffusion coefficients are: $0.016 \pm 0.004 \, \mu m^2/s$ (95% CI) for FB-mCh and $0.019 \pm 0.006 \, \mu m^2/s$ (95% CI) for Sun-GFP. Scale bars, 10 μm. Source data are provided as a Source Data file

Besides the HA frankenbody, there is only one other scFv capable of imaging single mRNA translation dynamics: the SunTag scFv[18,20,21,30,31]. Compared to the SunTag, the HA frankenbody binds it target epitope with lower affinity (nM versus pM). However, the SunTag epitope (EELLSKNYHLENE-VARLKK, 19 aa) is over twice the length of the HA epitope[23]. Furthermore, the relatively new SunTag is not as ubiquitous as the HA-tag, which has enjoyed widespread use in the biomedical sciences for over 30 years[35].

While the HA frankenbody and SunTag each have advantages, their combination creates a powerful genetically encoded toolset to quantify single mRNA translation dynamics in two colors. For example, their combination makes it possible to combine HA-epitopes and SunTag-epitopes in single mRNA reporters to visualize multiple open-reading frames[54,55] or create color gradients for multiplexed imaging. In this study, we combined the HA frankenbody with the SunTag to quantify the spatiotemporal dynamics of two mRNA species with different UTRs and ORFs. Unlike our earlier work with two mRNA species that sometimes co-localized during translation and that shared common UTRs, in this case, we did not observe co-localized translating mRNA. This would suggest that specific sequences within mRNAs likely dictate whether or not they co-localize during translation.

The genetic encodability of the HA frankenbody makes it a great value to researchers who have so far had to rely on expensive antibodies purified from hybridoma cells to label HA-tagged proteins. The HA frankenbody will therefore have an immediate and positive impact on the large group of researchers already employing the HA-tag in their studies. With the HA frankenbody, researchers can simply transfect cells or animals expressing HA-tagged proteins with DNA or mRNA encoding the frankenbody fused to a fluorescent protein. This enables the visualization and quantification of HA-tagged protein expression, localization, and dynamics in living systems, both at the single molecule level, as in our single-molecule tracking experiments, as well as across entire organisms, as in our zebrafish embryo experiments. While 1 × HA tags are adequate in both cases, we demonstrated fluorescence signal in zebrafish embryos can be

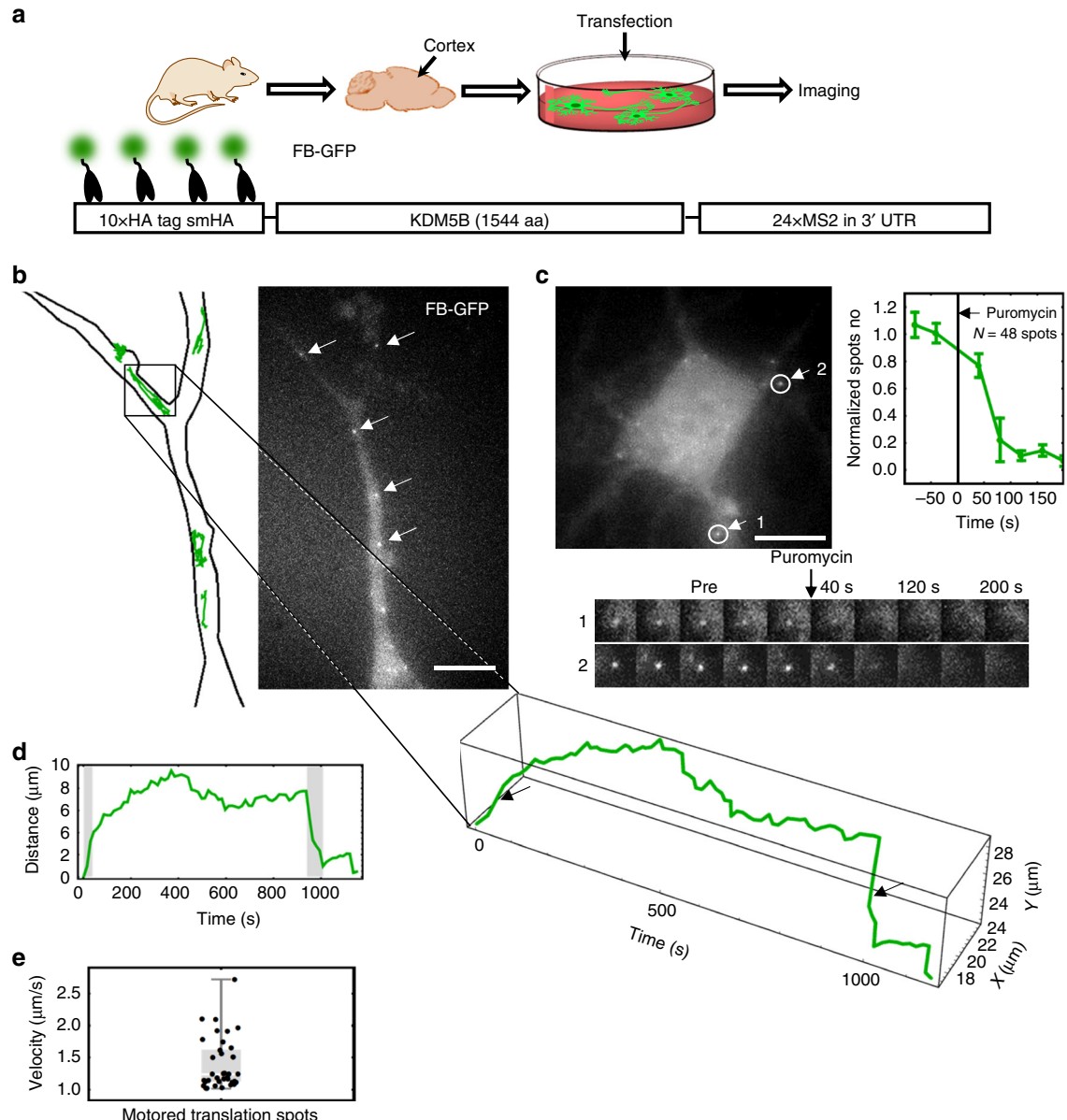

**Fig. 8** Monitoring local translation in living neurons. **a** A diagram depicting the preparation of rat primary cortical neurons for imaging. **b** The dendrite of a sample living neuron expressing frankenbody (FB-GFP) and the smHA-KDM5B-MS2 translation reporter ($n = 15$ cells in two independent experiments). White arrows indicate translation sites that were tracked, as depicted in the cartoon on the left. The spatiotemporal evolution of one mRNA track with directed motion is shown through time. **c** Puromycin treatment of a representative cell. Two circled translation sites were tracked as they disappeared following the addition of puromycin. Upper-right: The mean number of nascent chain spots normalized to pre-puromycin levels decreases after puromycin treatment ($N = 48$ spots from three cells in three independent experiments). Error bars, cell-to-cell SEM. **d** The travel distance through time for the translation spot highlighted in **b**. Gray highlights in **d** and black arrows in **b** indicate directed motion events. **e** Plot of velocities faster than $1\,\mu m/s$ (37 velocities from eight cells in two independent experiments). Mean: $1.40 \pm 0.07$ mm/s (spot-to-spot SEM). See also Supplementary Fig. 6. Scale bars, $10\,\mu m$. Source data are provided as a Source Data file. For the box and whisker plots, median is shown by a white line, the box indicates 25–75% range, and whiskers indicate 5–95% range

significantly amplified by simply repeating HA epitopes within tags (Fig. 9b,c, and Supplementary Fig. 8). This will be useful for the rapid and sensitive detection of short-lived HA-tagged proteins in vivo, similar to how the GFP-nanobody has been used to both amplify GFP fluorescence[13,14] and image the spatiotemporal dynamics of short-lived, GFP-tagged transcription factors during Drosophila development[5]. Finally, the frankenbody can be genetically fused to other protein motifs to create a wide array of tools for live-cell imaging or manipulation of HA-tagged proteins. It can also be co-evolved with the HA-epitope into other complementary probe/epitope pairs via directed evolution[56,57]. As

shown in Supplementary Fig. 1, the sequences of the two successful variants, $\chi_{15F11}^{HA}$ and $\chi_{2E2}^{HA}$, are slightly different, but their performance in living cells is almost identical. Thus, many positions of the frankenbody can be mutated without destroying its functionality, demonstrating potential for further evolution. We therefore anticipate the HA frankenbody will be a useful imaging reagent to complement the growing arsenal of live-cell antibody-based probes[23,58–60].

Given our success with CDR grafting, it should be relatively straightforward to generate more scFvs binding additional targets besides the HA epitope. However, the functionality of CDR

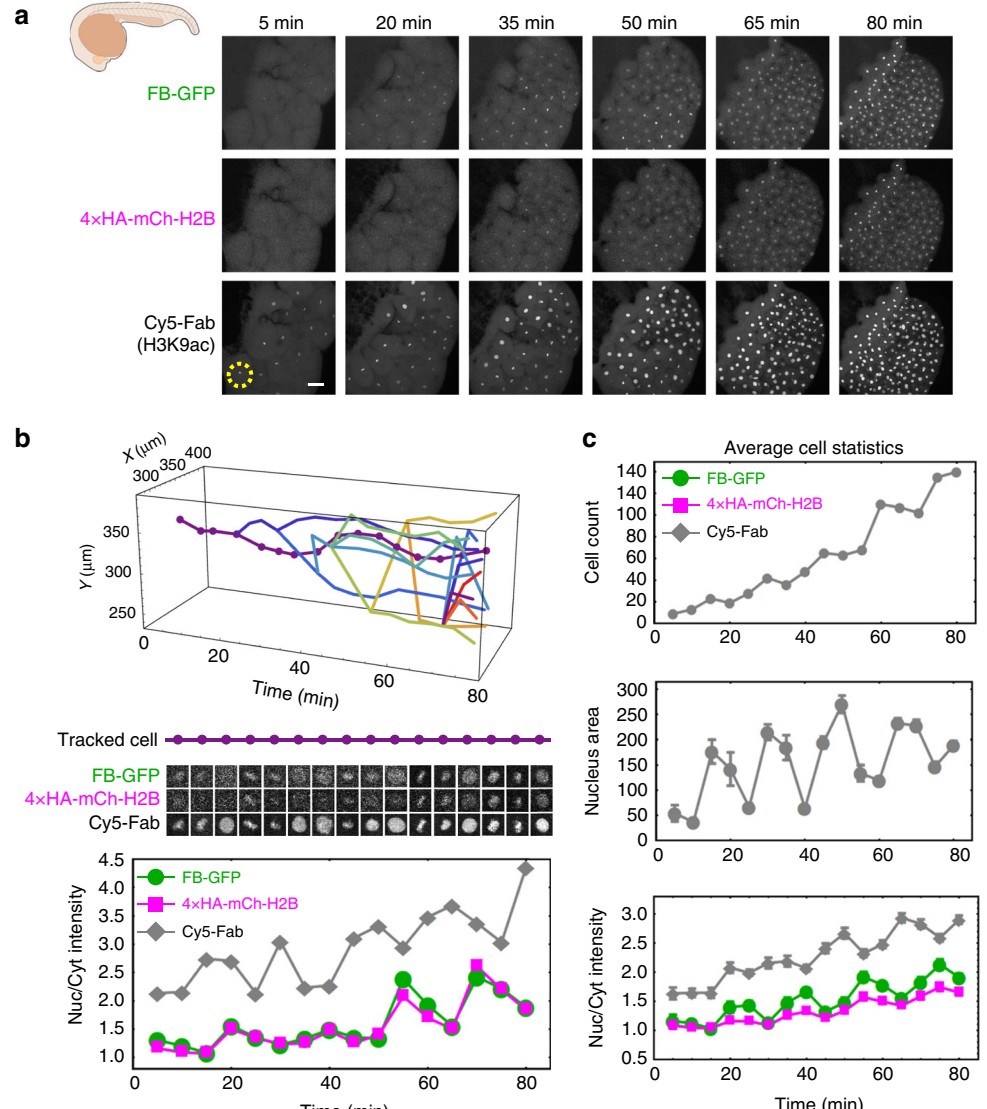

**Fig. 9** Monitoring HA-tagged proteins in zebrafish embryos. **a** Max-projection images from a zebrafish embryo with frankenbody (FB-GFP) and 4 × HA-mCh-H2B. Cy5-Fab labels nuclear histone acetylation as a positive control. **b** A nuclei (dash circle in **a**) and its progeny tracked in development. The nuclear to cytoplasmic ratio (Nuc/Cyt) from FB-GFP (green circles), HA-H2B-mCh (magenta squares), and Fab (gray diamonds) in the parental nuclei (dotted line). **c** Cell count (top), average nuclear area (units of pixel[2] with one pixel = 662 nm, middle) and Nuc/Cyt ratios for all tracked nuclei. See also Supplementary Figs. 7, 8 and Supplementary Movies 10, 11. Two embryos in two independent experiments. Error bars, SEM. Scale bar, 50 mm. Source data are provided as a Source Data file

grafted scFvs is still difficult to predict[33], so it remains unclear how generalizable the method is. According to our initial screening of scaffolds for frankenbodies, scFvs that have similar sequences will generally have a higher chance of being compatible grafting partners. We are therefore optimistic that as the price of determining antibody sequences continues to decrease, more scFvs will be constructed, tested, and verified to fold and function in living cells. The availability of compatible grafting partners will therefore increase, meaning less effort will be required to generate additional frankenbodies. Besides scFvs, another promising option is the continued development of nanobodies. For example, we recently were made aware of a nanobody capable of binding a repeated 15 aa epitope co-transaltionally (coined the MoonTag). Ultimately, as more and more genetically encoded epitope binders become available, we envision a panel of such probes will allow researchers to better capture the full lifecycles of proteins in vivo with high spatiotemporal resolution.

## Methods

**Plasmid construction**. Each chimeric anti-HA scFv tested in this study was constructed by grafting six CDR loops of an anti-HA antibody 12CA5 onto each selected scFv scaffold. The $\chi^{HA}_{15F11}$ plasmid was constructed in two steps: (1) a CDR-loop grafted scFv gblock and a H4K20me1 mintbody 15F11 vector[38] linearized by EcoRI restriction sites were ligated via Gibson assembly (House prepared master mix); (2) the linker connecting the scFv and EGFP, as well as EGFP, was replaced by a gblock encoding a flexible $(G_4S) \times 5$ linker and the mEGFP by Gibson assembly through NotI restriction sites. For the other four chimeric scFv plasmids, each CDR-loop grafted scFv gblock was ligated into the $\chi^{HA}_{15F11}$ vector linearized by EcoRI restriction sites via Gibson assembly.

The target plasmid 1 × HA-mCh-H2B was constructed by replacing the sfGFP of an Addgene plasmid sfGFP-H2B (Plasmid # 56367) with a 1 × HA epitope tagged mCherry gblock in which a NotI restriction site was inserted between the 1 × HA epitope and the mCherry for the following cloning. The 4 × HA-mCh-H2B was constructed by replacing the 1 × HA tag of the 1 × HA-mCh-H2B construct with a 4 × HA-epitope gblock through AgeI and NotI restriction sites. The 10 × HA-mCh-H2B (smHA-mCh-H2B) was constructed by replacing the 1 × HA tag of the 1 × HA-mCh-H2B construct with the smHA amplified from an Addgene plasmid smHA-KDM5B-MS2 (Plasmid # 81085) using primers NZ-098 and 099

(All primer sequences are shown in Supplementary Table 1). The smHA-H2B was constructed by replacing the $1 \times HA$-mCh of the $1 \times HA$-mCh-H2B construct with smHA amplified from the plasmid smHA-KDM5B-MS2 using primers NZ-098 and 100.

Another target plasmid $4 \times HA$-mCh-β-actin was constructed by replacing the H2B of the $4 \times HA$-mCh-H2B with a β-actin amplicon via BglII and BamHI restriction sites. The β-actin amplicon was amplified from an Addgene plasmid smFlag-ActinB-MS2 (Plasmid # 81083) using primers NZ-073 and NZ-074.

The $4 \times HA$-mRuby-Kv2.1 plasmid was generated by first amplifying the $4 \times$ HA tag from $4 \times HA$-mCh-H2B using the primers NZ-105 and 106. Then the $4 \times$ HA amplicon was introduced into a published plasmid pCMV-mRuby2-Kv2.1[61] (a gift from Dr. Michael Tamkun), which had been linearized by a restriction digest with AgeI, using Gibson assembly. The smHA-Kv2.1 plasmid was generated by PCR amplification of the rat Kv2.1 coding sequence from pBK-Kv2.1[40] and subsequent ligation into AsiSI and PmeI restriction sites on the plasmid smHA-KDM5B-MS2 using primers Kv2.1-1 and 2.

The H2B-mCh-$1 \times$ HA was constructed by two steps: (1) replace the $1 \times$ HA tag of the $1 \times HA$-mCh-H2B with H2B through AgeI and NotI sites, in which the H2B was amplified from $1 \times HA$-mCh-H2B using the primers NZ-092 and 093; (2) Replace the H2B at the C-terminus of mCherry with a $1 \times$ HA tag through BglII and BamHI sites, in which the $1 \times$ HA tag was synthesized by overlapping PCR with primers NZ-094 and 095. The Mito-mCh-$1 \times$ HA was constructed by replacing the H2B in H2B-mCh-$1 \times$ HA with Mito gblock[23] through AgeI and NotI sites. The Mito-mCh-smHA was constructed by replacing the $1 \times$ HA tag of Mito-mCh-$1 \times$ HA with smHA amplified from the smHA-KDM5B-MS2 using primers NZ-096 and 097. The mCh-H2B was generated by replacing the sfGFP of the plasmid sfGFP-H2B with an mCherry gblock using Gibson assembly.

The plasmids FB-mCh, FB-Halo, FB-SNAP were built by replacing the mEGFP of $\chi^{HA}_{15F11}$ with mCherry, HaloTag, and SNAP-tag gblocks, respectively.

pET23b-FB-GFP, the plasmid for recombinant frankenbody expression and purification, was Gibson assembled with a FB-GFP amplicon and a published plasmid pET23b-Sso7d[62,63] linearized by NdeI and NotI restriction sites. The FB-GFP amplicon was amplified from $\chi^{HA}_{15F11}$ by PCR with primers NZ-075 and 077.

For the translation assay, the reporter construct smHA-KDM5B-MS2 and SunTag-kif18b were obtained from Addgene (Plasmid # 81085 and # 74928), and the SunTag scFv plasmid (Plasmid # 60907) was modified by removing the HA epitope encoded in the linker. The HA epitope was removed by site-directed mutagenesis with QuikChange Lightning (Agilent Technologies) per the manufacturer's instruction using primers HAout-1 and 2.

The gblocks were synthesized by Integrated DNA Technologies and the recombinant plasmids were sequence verified by Quintara Biosciences. The sequences of primers are shown in Supplementary Table 1. All plasmids used for imaging translation were prepared by NucleoBond Xtra Midi EF kit (Macherey-Nagel) with a final concentration about $1 \, mg \, mL^{-1}$.

**U2OS cell culture**. U2OS cells (ATCC HTB-96) were grown in an incubator at 37 °C, humidified, with 5% $CO_2$ in DMEM medium (Thermo Scientific) supplemented with 10% (v/v) fetal bovine serum (Altas Biologicals), 1 mM L-glutamine (Gibco) and 1% (v/v) penicillin–streptomycin (Gibco or Invitrogen).

**Transfection and bead loading**. For tracking protein localization, cells were plated into a 35 mm MatTek chamber (MatTek) 2 days before imaging and were transiently transfected with Lipofectamine™ LTX reagent with PLUS reagent (Invitrogen) according to the manufacturer's instruction 18–24 h prior to imaging.

For imaging translation with mRNA labeled by MCP-Halo protein, cells were plated into a 35 mm MatTek chamber (MatTek) the day before imaging. On the day of imaging, before bead loading, the medium in the MatTek chamber was changed to Opti-MEM (Thermo Scientific) with 10% fetal bovine serum. A mixture of plasmids (smHA-KDM5B-MS2/FB) and purified MCP-HaloTag protein[17] were bead loaded as previously described[6,17,26]. Briefly, after removing the Opti-medium from the MatTek chamber, 4 µL of a mixture of plasmids (1 µg each) and MCP-HaloTag (130 ng) in PBS was pipetted on top of the cells and ~106 µm glass beads (Sigma Aldrich) were evenly distribute on top. The chamber was then tapped firmly seven times, and Opti-medium was added back to the cells. 3 h post bead loading, the cells were stained in 1 mL of 0.2 nM of JF646-HaloTag ligand[48] diluted in phenol-red-free complete DMEM. After a 20 min incubation, the cells were washed three times in phenol-red-free complete DMEM to remove glass beads and unliganded dyes. The cells were then ready for imaging.

For imaging translation without mRNA labeling, the cells plated on MatTek chamber were transiently transfected with the plasmids needed, smHA-KDM5B-MS2/FB with or without SunTag-kif18b/Sun (with the HA epitope removed), using Lipofectamine™ LTX reagent with the PLUS reagent (Invitrogen) according to the manufacturer's instruction on the day of imaging. 3 h post transfection, the medium was changed to phenol-red-free complete DMEM. The cells were then ready for imaging.

**Purification of FB-GFP**. *E. coli* BL21 (DE3) pLysS cells transformed with pET23b-FB-GFP were grown at 37 °C to a density of OD600 0.6 in 2xYT medium containing Ampicillin (100 mg $L^{-1}$) and Chloramphenicol (25 mg $L^{-1}$) with shaking.

Isopropyl-β-D-thiogalactoside (IPTG) was added to induce protein expression at a final concentration of 0.4 mM, and the temperature was lowered to 18 °C. Cells were harvested after 16 h by centrifugation and resuspended in PBS buffer supplemented with 300 mM NaCl, protease inhibitors (ThermoFisher), 0.2 mM AEBSF (20 mL $L^{-1}$ culture) and lysed by sonication. Lysate was clarified by centrifugation. The supernatant was loaded onto two connected HisTrap HP 5 mL columns (GE Healthcare), washed and eluted by a linear gradient of 0–500 mM imidazole. The fractions containing the POI were pooled, concentrated using Amicon Ultra-15 30 kDa MWCO centrifugal filter unit (EMD Millipore) and loaded onto a size-exclusion HiLoad Superdex 200 PG column (GE healthcare) in HEPES-based buffer (25 mM HEPES pH 7.9, 12.5 mM $MgCl_2$, 100 mM KCl, 0.1 mM EDTA, 0.01% NP40, 10% glycerol, and 1 mM DTT). The fractions containing FB-GFP protein were collected, concentrated, and stored at −80 °C after flash freezing by liquid nitrogen.

**Immunostaining**. U2OS cells were transiently transfected with $4 \times HA$-mCh-H2B or $4 \times HA$-mCh-β-actin with Lipofectamine™ LTX reagent with the PLUS reagent (Invitrogen). 26 h post transfection, the cells were fixed with 4% paraformaldehyde (Electron Microscopy Sciences) for 10 min at room temperature, permeabilized in 1% Triton 100 in PBS, pH 7.4 for 20 min, blocked in Blocking One-P (Nacalai Tesque) for 20 min, and stained at 4 °C overnight with purified FB-GFP protein (0.5 µg $mL^{-1}$ in 10% blocking buffer). The next morning, the cells were washed with PBS, and the POI was imaged by an Olympus IX81 spinning disk confocal (CSU22 head) microscope using a ×100 oil immersion objective (NA 1.40) under the following conditions: 488 nm (0.77 mW; measured at the back focal plane of the objective; herein all laser power measurements correspond to the back focal plane of the objective) and 561 nm (0.42 mW) sequential imaging for 50-time points without delay, $2 \times 2$ spin rate, 100 ms exposure time. Images were acquired with a Photometrics Cascade II CCD camera using SlideBook software (Intelligent Imaging Innovations). The immunostaining images were generated by averaging 50-time point images for each channel by Fiji[64].

**Western blots**. U2OS cells were transiently transfected with $4 \times HA$-mCh-H2B or $4 \times HA$-mCh-β-actin with Lipofectamine™ LTX reagent with the PLUS reagent. 10 h post transfection, the cells were harvested and lysed in 120 µL of RIPA buffer with cOmplete Protease Inhibitor (Roche). 6.5 µL of each cell lysate was loaded on a NuPAGE™ 4–12% Bis–Tris protein gel (Invitrogen) and run for 60 min at 100 V and 25 min at 200 V. Proteins were transferred to a PVDF membrane (Invitrogen), blocked in blocking buffer (5% milk powder in 0.05% TBS-Tween 20) for 1 h, and stained overnight with either purified FB-GFP protein (0.5 µg $mL^{-1}$ in blocking buffer) or anti-HA parental antibody 12CA5 (Sigma-Aldrich Cat #11583816001; 2000-fold dilution with final concentration 0.5 µg $mL^{-1}$ in blocking buffer). For the primary antibody 12CA5, an additional incubation for 1 h with anti-mouse antibody/Alexa488 (Thermo Fisher Cat #A11001; 5000-fold dilution in blocking buffer) was done. The POI was detected from the GFP fluorescence for FB-GFP or Alexa 488 for anti-HA antibody using a Typhoon FLA 9500 (GE Healthcare Life Sciences) with the following conditions: excitation wavelength 473 nm, LPB filter (≥510 nm), 300 V photomultiplier tube and 10 µm pixel size. The uncropped and unprocessed western blots are supplied in Supplementary Fig. 3.

**Fluorescence recovery after photobleaching**. To study the binding affinity of FB to HA epitopes in living cells, FRAP experiments were performed on cells transiently transfected with $4 \times HA$-mCh-H2B (1.25 µg) and FB-GFP (1.25 µg) 24 h before FRAP. The images were acquired using an Olympus IX81 spinning disk confocal (CSU22 head) microscope coupled to a Phasor photomanipulation unit (Intelligent Imaging Innovations) with a ×100 oil immersion objective (NA 1.40). Before photobleaching, 20 or 10 frames were acquired with 1 or 5 s time interval. The images were captured using a 488 nm laser (0.77 mW) laser with 100 ms exposure time followed by 561 nm (0.42 mW) laser with 15 ms exposure time. The spinning disk was set up at $1 \times 1$ spin rate. After acquiring pre-FRAP images, the p488 nm laser (from the Phasor unit for photobleaching) at 17 mW with 100 ms exposure time, or 4 mW with 500 ms exposure, was used to photobleach a circular region in the nucleus. After photobleaching, 30 images were captured without delay or with 500 ms delay, and then 100 images with a 1 s delay and 100 images with a 5 s delay were acquired using the same imaging settings as the pre-FRAP images. The fluorescent intensity through time of the photobleached spot were exported using the Slidebook software. The fluorescent intensity of the nucleus and background were obtained by Fiji[64] after correcting for cell movement using the StackReg Fiji plugin[65]. The FRAP curve and $t_{half}$ were obtained using easyFRAP-web[66], according to the website instructions. Figure 4b, d were generated by *Mathematica* (Wolfram Research).

**MCP purification**. MCP-HaloTag was purified by immobilized metal affinity chromatography[17]. Briefly, the His-tagged MCP-HaloTag was purified through a Ni-NTA-agarose (Qiagen) packed column per the manufacturer's instructions, with minor modifications. *E. coli* expressing the interested protein was lysed in a PBS buffer with a complete set of protease inhibitors (Roche) and 10 mM imidazole. The resin was washed with PBS-based buffer containing 20 and 50 mM imidazole. The protein was then eluted in a PBS buffer with 300 mM imidazole.

The eluted His-tagged MCP was dialyzed in a HEPES-based buffer (10% glycerol, 25 mM HEPES pH 7.9, 12.5 mM MgCl$_2$, 100 mM KCl, 0.1 mM EDTA, 0.01% NP-40 detergent, and 1 mM DTT), snap-frozen in liquid nitrogen, and then stored at −80 °C.

**Cell preparation for nascent chain tracking.** The reporter plasmid smHA-KDM5B-MS2 (Addgene plasmid #81085) and the FB construct were either transiently transfected without the MCP-HaloTag protein or bead loaded with MCP-HaloTag protein into U2OS cells plated on a 35 mm MatTek chambers 4–6 h before imaging. 3 h later, if MCP-HaloTag protein was bead loaded, the cells were stained with the JF646-HaloTag ligand, then washed with phenol-red-free complete DMEM medium. If no MCP-HaloTag protein was needed, the medium of the cells was changed to phenol-red-free complete DMEM medium 3 h post transfection. The cells were then ready for imaging.

For multiplexed imaging, two reporter plasmids, smHA-KDM5B-MS2 and SunTag-kif18b, as well as two probes, FB-mCh and Sun-GFP (with the HA epitope removed), were transiently transfected into U2OS cells plated on a MatTek chamber 4–6 h before imaging. 3 h later, the medium of the cells was changed to phenol-red-free complete DMEM medium. The cells were then ready for imaging.

**Imaging condition for translation and colocalization assays.** To image single mRNAs and their translation status with FB, a custom-built widefield fluorescence microscope based on an inclined illumination (HILO) scheme was used[17,67]. Briefly, the excitation beams, 488, 561, 637 nm solid-state lasers (Vortran), were coupled and focused off-axis on the rear focal plane of the objective lens (APON 60XOTIRF, Olympus). All reported laser powers were measured at the back-focal plane of the objective. The emission signals were split by an imaging grade, ultra-flat dichroic mirror (T660lpxr, Chroma). The longer emission signals (far-red) after splitting were passed through a bandpass filter (FF01-731/137-25, Semrock). The shorter emission signals (red and green) after splitting were passed through either a bandpass filter for red (FF01-593/46-25, Semrock) or a bandpass filter for green (FF01-510/42-25, Semrock) installed in a filter wheel (HS-625 HSFW TTL, Finger Lakes Instrumentation). The longer (far-red) and the shorter (red and green) emission signals were detected by separate two EM-CCD cameras (iXon Ultra 888, Andor) by focusing with a 300 mm achromatic doublet lenses (AC254-300-A-ML, Thorlabs). The combination of ×60 objective lens from Olympus, 300 mm tube lens, and iXon Ultra 888 produces ×100 images with 130 nm pixel$^{-1}$. A stage top incubator for temperature (37 °C), humidity, and 5% CO$_2$ (Okolab) is equipped on a piezoelectric stage (PZU-2150, Applied Scientific Instrumentation) for live cell imaging. The lasers, the cameras, the piezoelectric stage, and the filter wheel were synchronized by an open source microcontroller, Arduino Mega board (Arduino). Imaging acquisition was performed using open source Micro-Manager software (1.4.22)[68].

The imaging size was set to the center 512 × 512 pixels$^2$ (66.6 × 66.6 μm$^2$), and the camera integration time was set to 53.64 ms. The readout time of the cameras from the combination of our imaging size, readout mode (30 MHz), and vertical shift speed (1.13 μs) was 23.36 ms, resulting in our imaging rate of 13 Hz (70 ms per image). Red and green signals were imaged alternatively. The emission filter position was changed during the camera readout time. To minimize the bleed-through, the far-red signal was simultaneously imaged with the green signal. To capture the whole thickness of cells, 13 z-stacks with a step size of 500 nm (6 μm in total) were imaged using the piezoelectric stage. This resulted in our total cellular imaging rate of 1 Hz for imaging either red or green signals, and 0.5 Hz for imaging both red and green signals regardless of far-red imaging.

For Figs. 1c, d, 2a, e, a single plane of the cells was imaged continuously at 6.5 Hz for 100 time points and averaged throughout the time (lasers: 488 nm, 130 μW; 561 nm, 4 μW (Fig. 1c), 90 μW (Fig. 1d), 19 μW (Fig. 2a), 155 μW (Fig. 2e); 637 nm, 220 μW). For Fig. 2b (left), the cell was imaged at 0.5 Hz with 13 z-stacks every time point and averaged throughout the time. The acquired averaged 13 z-stacks were deconvolved using Fiji. For Fig. 6b, c, e and f, cells were imaged every 10 s with 13 z-stacks per time point (lasers: 488 nm, 13 μW (Fig. 6b), 18 μW (Fig. 6c); 561 nm, 172 μW (Fig. 6f); 637 nm, 150 μW (Fig. 6b, c), 35 μW (Fig. 6e)). For Fig. 7b, the cell was imaged continuously at 0.5 Hz with 13 z-stacks every time point (lasers: 488 nm, 130 μW; 561 nm, 172 μW). For Fig. 8b, the cells were imaged every 14 s with 13 z-stacks every time point (laser: 488 nm, 70 μW). For Fig. 8c, cells were imaged every 40 s with 13 z-stacks every time point (laser: 488 nm, 130 μW). For motored translation spots velocity determination (Fig. 8e), the neurons were imaged continuously at 1 Hz with 13 z-stacks every time point.

For Fig. 2b (right) and 2c, the co-localization was imaged by the Olympus IX81 spinning disk confocal (CSU22 head) microscope described before using a ×100 oil immersion objective (NA 1.40) under the following conditions: 488 nm (0.77 mW) and 561 nm (0.42 mW) sequential imaging for five time points without delay with multiple z slices to cover the whole cell body for each time point, 1 × 1 spin rate, exposure time adjusted by cell brightness. Images were acquired with a Photometrics Cascade II CCD camera using SlideBook software (Intelligent Imaging Innovations). The displayed images in figures were generated by averaging five time points and then a max-projection of all z-slices was performed by Fiji[64].

**Particle tracking of translation sites.** Single translation site detection and tracking was performed on maximum intensity projection images with custom *Mathematica* (Wolfram Research) code[17]. Briefly, the images were processed with a bandpass filter to highlight particles, and then binarized to detect their intensity-centroids as positions using the built-in Mathematica routine ComponentMeasurements. Detected particles were tracked and linked through time via a nearest neighbor search. The precise coordinates (super-resolved locations) of mRNAs and translation sites were determined by fitting (using the built-in Mathematica routine NonlinearModelFit) the original images to 2D Gaussians of the following form:

$$I(x,y) = I_{BG} + I e^{-\frac{(x-x_0)^2}{2\sigma_x^2} - \frac{(y-y_0)^2}{2\sigma_y^2}} \qquad (1)$$

where $I_{BG}$ is the background fluorescence, $I$ the particle intensity, $(x_0, y_0)$ the particle location, and $(\sigma_x, \sigma_y)$ spreads of the particle. The offset between the two cameras was registered using the built-in Mathematica routine FindGeometricTransform to find the transform function that best aligned the fitted positions of 100 nm diameter Tetraspeck beads evenly spread out across the image field-of-view.

For Figs. 6c, e, f, 7c, 8b, d, particles were tracked by custom *Mathematica* code and further plotted with *Mathematica*. For Fig. 7c, average mean squared displacements were calculated from the Gaussian-fitted coordinates (from 2D maximum intensity projection images). The diffusion constant was obtained by fitting the first five time points to a line with slope $m = 4D$, where $D$ is the diffusion coefficient. The single motored translation spots (Fig. 8e) were tracked by the Fiji plugin TrackMate[69] after max-projection and further plotted with *Mathematica*. All *Mathematica* code is available upon request.

**Single molecule tracking of 1×HA-tagged proteins in cells.** The day before imaging, cells were plated on MatTek chamber and transiently transfected with the 1 × HA-mCh-H2B and FB-Halo using Lipofectamine$^{TM}$ LTX reagent with the PLUS reagent (Invitrogen) according to the manufacturer's instruction. 3 h post transfection, the medium was changed to complete DMEM. Before imaging, the cells were stained with sodium borohydride (NaBH$_4$) treated Halo ligand TMR (Promega)[45]. Briefly, 1 μL of 1 mM Halo ligand TMR dye was reduced for 10 min in 200 μL of 50 mM NaBH$_4$ solution (pre-dissolved in PBS for 10 min, pH 7.4). Next, 200 μL of the reduced TMR was diluted with 800 μL of phenol-red-free DMEM to produce 1 mL reduced-TMR media. Media from transfected cells was replaced by the reduced-TMR media and cells were then placed in an incubator (5% CO$_2$, 37 °C) for 30 min for staining. The cells were then washed three times with phenol-red-free DMEM. Between washes, cells were incubated for 5 min in an incubator (5% CO$_2$, 37 °C).

The cells were imaged using a custom-built widefield fluorescence microscope based on an inclined illumination (HILO) scheme[17,67]. The imaging field-of-view was set to 256 × 256 pixels$^2$ (33.3 × 33.3 μm$^2$), and the camera integration time was set to 30 ms. The cells were imaged with a 7.7 mW 561 nm laser at an imaging rate of 43.8 ms per image for a total of 10,000 time points (7.3 min). During imaging, a 6.2 mW 405 nm laser was pulsed on for 50 ms once every 10 s to photoactivate the Halo-TMR reduced ligand. Single molecules were tracked using the Fiji plugin TrackMate[69]. To ensure tracks represent FB-Halo bound to 1 × HA-mCh-H2B, tracks were further filtered in *Mathematica*. The filter eliminated tracks of length less than 16 frames. Further, all jumps between frames had to be less than 220 nm. This criteria has been used by others to distinguish transcription factors that are chromatin-bound from those that are unbound[46]. Finally, in *Mathematica*, tracks were color-coded either according to the time at which they were acquired (as in Supplementary Fig. 5) or the average jump size between frames (as in Fig. 5).

**Puromycin treatment.** U2OS cells transiently transfected with smHA-KDM5B-MS2 and FB or bead loaded with smHA-KDM5B-MS2, FB and MCP-HaloTag were imaged as above with 10 s intervals between frames. After acquiring 5 or 10 time points as pre-treatment images, cells were treated with a final concentration of 0.1 mg mL$^{-1}$ puromycin right before acquiring the 6th or 11th time point. After puromycin was added, the cells were imaged under the same conditions used for the pre-treatment imaging until the translation spots disappeared.

**Neuron culture and transfection.** Rat cortical neurons were obtained from the discarded cortices of embryonic day (E)18 fetuses which were previously dissected to obtain the hippocampus, and frozen in Neurobasal medium (ThermoFisher Scientific) containing 10% fetal bovine serum (FBS, Atlas Biologicals) and 10% DimethylSulfoxide (Sigma-Aldrich, D8418) in liquid nitrogen. Cryopreserved rat cortical neurons were plated at a density of ~15,000–30,000 cells per cm$^2$ on MatTek dishes (MatTek) and cultured in Neurobasal medium containing 2% B27 supplement (ThermoFisher Scientific), 2 mM L-alanine/L-glutamine and 1% FBS (Atlas Biologicals). Transfections were performed after 5–7 days in culture by using Lipofectamine 2000 (ThermoFisher Scientific) according to the manufacturer's instructions. Neurons co-expressing 4 × HA-mRuby-Kv2.1 and FB-GFP were imaged 1–2 days post-transfection (Fig. 2b). Neurons co-expressing smHA-Kv2.1 and FB-GFP were imaged 1–7 days post-transfection (Supplementary Fig. 2 left). For translation assays, neurons were imaged 4–12 h post-transfection. All neuron imaging experiments were carried out in a temperature-controlled (37 °C), humidified, 5% CO$_2$ environment in

Neurobasal medium without phenol red (ThermoFisher Scientific). Neuronal identity was confirmed by following processes emanating from the cell body to be imaged for hundreds of microns to ensure they were true neurites.

**Monitoring Zebrafish development.** All zebrafish experiments have been approved by the Tokyo Tech Genetic Experiment Safety Committee (I2018001) and animal handling is operated according to the guidelines. To visualize FB-GFP in zebrafish embryo, mRNAs for FB-GFP and N × HA-mCh-H2B were prepared. DNA fragments coding FB-GFP and N × HA-mCh-H2B were inserted into a plasmid containing the T7 promoter and poly A[70]. The subsequent plasmids (T7-FB-GFP and T7-N × HA-mCh-H2B) were linearized with the XbaI restriction site for in vitro transcription using mMESSAGE mMACHINE kit (ThermoFisher Scientific). RNA was purified using RNeasy Mini Elute Cleanup Kit (QIAGEN) and resuspended in water. Before microinjection, zebrafish (AB) eggs were dechorionated by soaking in 2 mg mL$^{-1}$ pronase (Sigma Aldrich; P5147) in 0.03% sea salt for 10 min. A mixture (~0.5 nL) containing mRNA (200 pg each for FB-GFP and N × HA-mCh-H2B) was injected into the yolk (near the cell part) of 1-cell stage embryos. For a negative control, HA-mCh-H2B mRNA was omitted. 5–10 min after mRNA injection, Cy5-labeled Fab specific to endogenous histone H3 Lys9 acetylation (CMA310)[25] was injected (100 pg in ~0.5 nL). Injected embryos were incubated at 28 °C until the four-cell stage and embedded in 0.5% agarose (Sigma Aldrich, A0701) in 0.03% sea salt with the animal pole down on a 35-mm glass bottom dish (MatTek). The fluorescence images were collected using a confocal microscope (Olympus; FV1000) equipped with a heated stage (Tokai Hit) set at 28 °C and a UPLSAPO ×30 silicone oil immersion lens (NA 1.05), operated by the built-in FV1000 software FLUOVIEW ver.4.2. Three color images were sequentially acquired every 5 min using 488, 543, and 633 nm lasers (640 × 640 pixels; 0.662 μm pixel$^{-1}$, pinhole 800 μm; 2.0 μs pixel$^{-1}$) without averaging. Maximum intensity projections were created from 20 z-stacks with 5 μm.

Nuclei within zebrafish embryos were tracked in 4D using the Fiji plugin TrackMate[69]. Results were post-processed and plotted with *Mathematica*. To quantify the number and area of nuclei and the average nuclear, cytoplasmic, and nuclear:cytoplasmic intensity through time, the intensity of all nuclei in maximum intensity projections was measured using the built-in *Mathematica* function *ComponentMeasurements*. *ComponentMeasurements* requires binary masks of the objects to be measured. Binary masks of the nuclei were made using the built-in Mathematica function *Binarize* with an appropriate intensity threshold to highlight just nuclei in images from Cy5-labeled Fab (specific to endogenous histone H3 Lys9 acetylation). Masks of the cytoplasm around each nuclei were made by dilating the nuclear masks by 4 pixels (using the built-in command *Dilation*) and then subtracting from the dilated mask the original nuclear masks dilated by 1 pixel. This creates ring-like masks around each nuclei, from which the average cytoplasmic intensity was measured.

**Surface plasmon resonance.** Binding kinetics of purified FB to the HA epitope tag was measured by surface plasmon resonance (OpenSPR, Nicoyalife). After biotin-labeled HA peptide was captured by a Streptavidin sensor chip (Nicoyalife), diluted purified FB-GFP in PBS running buffer, pH 7.4, was slowly flowed over the sensor chip for 5 min to allow interaction. The running buffer was then allowed to flow for 10 min to collect the dissociation data. The non-specific-binding curve was obtained by flowing the same concentration of FB-GFP in the same running buffer over a different Streptavidin sensor chip (Nicoyalife). The data from the control was collected exactly as in the experiment. For 100 and 30 nM of FB-GFP, the binding response was significantly higher than the control, with negligible non-specific-binding interaction. Therefore, we chose those two concentrations for binding kinetics fitting. After subtracting the control, the signal response vs. time curve was obtained, as shown in Supplementary Fig. 4. Binding kinetic parameters were obtained by fitting the curve to a one-to-one binding model using Trace-Drawer (Nicoyalife) software (Supplementary Fig. 4).

**Reporting summary.** Further information on research design is available in the Nature Research Reporting Summary linked to this article.

## Data availability

The source data underlying Figs. 1e, 2a, d, f, 4b, d, 5b, c, 6c, e, f, 7c, 8b–e, 9b, c and Supplementary Figs. 4–7c and 8 are provided as a Source Data file. The raw images were deposited on figshare and can be accessed at https://doi.org/10.6084/m9.figshare.8072438. All other data are included in the manuscript and/or supplemental materials are available from the corresponding authors upon reasonable request.

## Code availability

All *mathematica* codes included in the manuscript or supplementary materials are available from the authors upon request.

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

## Acknowledgements

We thank all members of the Stasevich lab for input and helpful suggestions, especially Kenneth Lyon for removing the HA epitopes from the Sun-GFP construct (Addgene #60907) and Lara Perinet for helping with plasmid preparation. We also thank members of the CSU TagTeam; in particular, Dr. Chris Snow, Dr. Brian Geiss, and Steve Foster, for valuable discussions. Finally, we thank Laurie S. Minamide, Dr. James R. Bamburg and Dr. Michael M. Tamkun of CSU for supplying neurons, Dr. Michael M. Tamkun of CSU for supplying pCMV-mRuby-Kv2.1 plasmid and Dr. Luke Lavis of Janelia for supplying JF646 dye in this study. The research reported in this publication was supported by Colorado State University's Office of the Vice President for Research Catalyst for Innovative Partnerships Program. The content is solely the responsibility of the authors and does not necessarily represent the official views of the Office of the Vice President for Research. This work was also funded through an award to T.J.S. by the NIH (R35GM119728). T.J.S. is also supported by funds from the Boettcher Foundation's Webb-Waring Biomedical Research Program. H.K. was supported by KAKENHI JP18H05527.

## Author contributions

Conceptualization, N.Z. and T.J.S.; Methodology, N.Z., T.M., P.D.F., K.K., H.O., Y.S., H.K., and T.J.S.; Software, T.M. and T.J.S.; Validation, N.Z., P.D.F., K.K., and H.K.; Formal analysis, N.Z. and T.J.S.; Investigation, N.Z., T.M., P.D.F., K.K., H.O., Y.S., H.K., and T.J.S.; Resources, N.Z., H.K., and T.J.S.; Data curation, N.Z. and T.J.S.; Writing-original draft, N.Z. and T.J.S.; Writing-review & editing, N.Z. and T.J.S.; Visualization, N.Z., P.D.F., T.J.S., K.K., H.O., and H.K.; Supervision, T.J.S. and H.K.; Project Administration, N.Z. and T.J.S.; Funding acquisition, T.J.S. and H.K.

## Additional information

**Competing interests:** The authors declare no competing interests.

