## [Peer Review File · Nature Communications]

Reviewers' comments:

Reviewer #1 (Remarks to the Author):

Stasevich and coworkers describe the engineering of anti-HA scFv antibody to be functionally working in the intracellular environment and its uses as probe by DNA transfection for imaging HA-tagged protein and tracking single mRNA translation dynamics in living cells. The expression of scFvs within the reduced cytoplasm of living cells is generally limited by their instability and insolubility. To generate intracellularly functional anti-HA scFv, the authors grafted all six CDRs from anti-HA 12CA5 into the corresponding positions of five scFv scaffolds that were previously reported to be functional inside of cells, screened them in living cells and isolated two scFvs colocalizing with HA-tagged proteins, and finally chose one functional anti-HA scFv (termed HA frankenbody (FB)). Then, the authors co-transfected two expression plasmids: One encoding GFP-fused FB (FB-GFP) and 4xHA- and mCherry-tagged protein of interest (POI). As the HA peptides are translated and emerge from the ribosome exit tunnel, the tags are rapidly bound by the soluble and already fluorescent FB-GFP, thereby the authors can analyze the colocalization to chase the location of POI and its mRNA translation dynamics.

This kind of approach (i.e., using genetically encoded antibody probe for imaging of POI and its single mRNA) described in this manuscript has been reported in previous works including GFP-fused anti-SunTag scFv (Sun-GFP) and GFP-fused POI-binding scFvs, some of them were cited in the manuscript. Only a new thing in this manuscript is generating anti-HA scFv that can be expressed functionally to specifically bind to HA tag inside of cells. From antibody engineering point of view, engineering and generation of scFvs that are functional with proper folding and stability in the cytosol of living cells has been widely reported including intrabody by various approaches. Accordingly, despite the interesting results, this study significantly lacks the conceptual advance to be published in Nature Communications.

The followings are some specific comments.

1. Most critically, this manuscript lacks clear statistical analysis. There are no information on statistical analyses of data in Figure legends and M&M. The authors should describe how many times they repeat the experiments and, if possible, some images and data should be quantified.

2. The affinity of FB to HA tag is 13 nM, which is relatively low. Rather than just CDR grafting into the stable framework of 15F11 antibody, the following engineering (including optimization of framework residues (e.g., Vernier zone residues influencing CDR conformation)) might result in higher affinity anti-HA FB.

Due to the little bit lower affinity, the authors seemed to fuse 4x HA to POI to increase affinity by avidity effect. However, the long repeating HA tag might influence the conformation, folding, and/or localization of POI. Even, 10x HA encoding mRNA at the N-terminal might affect the translation efficiency and mRNA stability. The long HA tag could potentially exert appreciable perturbation, which can mislead the data. This reviewer wonders whether current FB-GFP can also efficiently colocalize 1x or 2x HA fused POI or mRNA. Further this reviewer wonder whether HA tag can be fused at the C-terminus of POI and 3'-UTR of mRNA. These results can validate the current results.

3. Fig. 2a: The authors stated that "frankenbodies can pass through the nuclear pore and bind target nuclear proteins". FB-GFP does not have any nucleus localization signal. How FB-GFP can enter nucleus? Based on the Fig. 2a data, FB-GFP binds to H2B in the cytosol, the complex of which then enters nucleus owing to H2B.

4. Fig. 2d: The authors stated that "Encouragingly, all three frankenbody constructs colocalized with HA-tagged H2B in the nucleus of living U2OS cells, similar to the original GFP-tagged frankenbody (Fig. 2D, upper)." However, the results show that some portions of FB-Halo and FB-mCh remain in the cytosol.

5. Fig. 2c: For a model of membrane protein (Kv2.1), the image does not clearly support the peripheral membrane localization. More sharp z-axis image is desirable or other transmembrane protein might be suitable as a model membrane protein.

Reviewer #2 (Remarks to the Author):

Report on manuscript entitled "A genetically encoded probe for imaging HA-tagged protein translation, localization, and dynamics in living cells and animal", by Ning Zhao et al.

In this paper, the authors report the development of a monochain antibody against the HA tag (FB), which works in an intracellular environment. The antibody has an affinity in the 10 nM range in vitro, and an off rate of 2-3 minutes in vivo. It works not only as a GFP fusion, but also when fused to mCherry, SNAP and HaloTag. When the protein of interest is fused to a repeated HA epitope, such as the spaghetti monster that contains 10xHA tag, the single chain antibody allows to visualize translation sites in living cell, as was previously reported for the SunTag. Finally, it is also suitable for labelling proteins in live zebrafish embryos.

Overall, the authors present a nice and useful tool to image translation and single molecule of proteins. There has been a strong need for good tools to label peptide tag in live cells, and this responds to this need. It is also highly complementary to the SunTag system that has been heavily used since its publication. The study is straightforward and well documented, and I strongly support publication in Nat Comm.

The following comments should however be addressed:

1-Although the scFv is soluble with the tested proteins, the author should run a more demanding aggregation test, for instance using a 10xHA-mitochondrial protein, as done with the original SunTag paper. The absence of mitochondrial clustering would be a strong argument that the system does not lead to artifactual aggregation at high concentration.

2-Figure 2c: the images of FB-GFP+smHA-Kv2.1 and GFP-Kv2.1 look quite different. Why not use a smHA-mCherry-Kv2.1 as in the other panels ?

3-Figure 2D: the efficiency of labelling seem to vary for the different constructs. Is this real or just an impression from the cells that were chosen in the images ? if FB-GFP and FB-HaloTag are co-expressed, is the labelling similar for both fusions ?

4-Figure 3b: I think that it would be a nice addition to the paper if the authors would use FB-HaloTag as a probe in fixed cells and demonstrate that this is allows to perform STORM-type of super-resolution imaging.

5-Figure 3C: the blot on the right is overexposed and should be replaced.

6-Figure 8. It is unclear from what is shown that the FB-GFP provides any advantage over the mCherry label fused to H2B. Can this experiment be repeated with for instance a 10xHA-sm ? It would then perhaps highlight a strong gain in sensitivity of the FB approach.

7-In the discussion of motorized transport of polysome, the author could also cite ref21, which also report this phenomenon.

Reviewers' comments:

Reviewer #1 (Remarks to the Author):

Stasevich and coworkers describe the engineering of anti-HA scFv antibody to be functionally working in the intracellular environment and its uses as probe by DNA transfection for imaging HA-tagged protein and tracking single mRNA translation dynamics in living cells. The expression of scFvs within the reduced cytoplasm of living cells is generally limited by their instability and insolubility. To generate intracellularly functional anti-HA scFv, the authors grafted all six CDRs from anti-HA 12CA5 into the corresponding positions of five scFv scaffolds that were previously reported to be functional inside of cells, screened them in living cells and isolated two scFvs colocalizing with HA-tagged proteins, and finally chose one functional anti-HA scFv (termed HA frankenbody (FB)). Then, the authors co-transfected two expression plasmids: One encoding GFP-fused FB (FB-GFP) and 4xHA- and mCherry-tagged protein of interest (POI). As the HA peptides are translated and emerge from the ribosome exit tunnel, the tags are rapidly bound by the soluble and already fluorescent FB-GFP, thereby the authors can analyze the colocalization to chase the location of POI and its mRNA translation dynamics.

We thank the reviewer for the comments and feedback. Below we provide a point-by-point rebuttal. To facilitate the review, all of our responses and changes are shown with a dark-purple font.

This kind of approach (i.e., using genetically encoded antibody probe for imaging of POI and its single mRNA) described in this manuscript has been reported in previous works including GFP-fused anti-SunTag scFv (Sun-GFP) and GFP-fused POI-binding scFvs, some of them were cited in the manuscript. Only a new thing in this manuscript is generating anti-HA scFv that can be expressed functionally to specifically bind to HA tag inside of cells. From antibody engineering point of view, engineering and generation of scFvs that are functional with proper folding and stability in the cytosol of living cells has been widely reported including intrabody by various approaches. Accordingly, despite the interesting results, this study significantly lacks the conceptual advance to be published in Nature Communications.

We feel the main novelty of the HA frankenbody is its remarkable versatility. In the revised manuscript, we demonstrate the frankenbody can be used to image mature proteins with 1x (new) to 10x N/C-terminal (new) HA tags in a variety of colors and cellular environments. In all cases we have tested so far, including cancer cell lines (U2OS), sensitive primary neurons, and full developing model organisms (zebrafish embryos), the HA frankenbody performs exceedingly well. We also now demonstrate the HA frankenbody can be used to track single 1xHA-tagged proteins (Fig. 5). Such diverse applications, in combination with the fact that the HA epitope is already widely

used by the community, makes us confident the HA frankenbody will immediately have a big impact on the field of live-cell imaging.

Adding to the versatility of the HA frankenbody is its ability to image nascent peptide chains co-translationally, not possible with any other system besides the SunTag. The frankenbody therefore fills a significant need in the field, enabling two-color comparative analyses of single mRNA translation dynamics in living cells (as demonstrated in Fig. 7). This will help usher in a new field of mRNA translation research, allowing, for example, scientists to fairly compare in the model system of their choice the translation dynamics of a wild-type and a mutant mRNA with unprecedented spatio-temporal resolution.

Finally, although the engineering of the HA frankenbody does not represent a conceptual advance in and of itself, we point out that the final sequence of the frankenbody could not have been anticipated or predicted by any algorithm we are aware of. Thus, the combination of the 15F11/2E2 scaffold and anti-HA CDRs is inherently novel. From the outset it was not clear the unstructured HA epitope would be large enough to bind an scFv with sufficient affinity and specificity for live-cell imaging. For example, the SunTag epitope is 19 aa in length and has secondary structure to facilitate binding. Likewise, the BC2 epitope, which is bound by a bivalent nanobody, is 15 aa in length (and it remains unclear if translation imaging is possible with this system). The HA epitope is therefore the smallest epitope that has been used so far for live-cell imaging.

In summary, we have demonstrated the HA frankenbody can be used to image both mature and nascent protein dynamics with single molecule resolution in a wide range of experimental settings. Given this extreme versatility, we believe our manuscript will be of great interest to the broad readership of Nature Communications. We now add a paragraph in the introduction to further highlight the versatility of the HA frankenbody (page 5).

The followings are some specific comments.

1. Most critically, this manuscript lacks clear statistical analysis. There are no information on statistical analyses of data in Figure legends and M&M. The authors should describe how many times they repeat the experiments and, if possible, some images and data should be quantified.

We now include statistical analyses to back up all of the major claims in our manuscript. In particular, all figure legends now state the number of cells and independent experiments that were performed. We also add in the following specific analyses:

- (1) Fig. 1f box-whisker plot with p-values to demonstrate the superiority of the 15F11 scaffold for live-cell imaging of HA epitopes.
- (2) Figs. 2a,d,f box-whisker plots with p-values to quantify experiments with 1x tags on the N/C terminus, to quantify imaging signal-to-noise, and to compare different fluorophores on the HA frankenbody, respectively.

- (3) Fig. 4d to quantify HA frankenbody FRAP recovery half times.
- (4) Fig. 5 and Sup. S4 to quantify single molecule tracking experiments.
- (5) Fig. 6c shows the average puromycin induced run-off with error bars representing cell-to-cell SEM. Fig. 6e,f show the puromycin induced run-off curve from the displayed cell.
- (6) Fig 7c, we've corrected the plot and fit to include data from all 8 cells (before we accidentally only used data from 2 cells).
- (7) Fig. 8c shows the average response to puromycin from three neurons.
- (8) Fig. 8e and S5 to quantify the distribution of motored transcript velocities observed in neurons.
- (9) Fig. S7 to directly compare 1x, 4x, and 10x HA-tagged constructs in live zebrafish embryos.

2. The affinity of FB to HA tag is 13 nM, which is relatively low. Rather than just CDR grafting into the stable framework of 15F11 antibody, the following engineering (including optimization of framework residues (e.g., Vernier zone residues influencing CDR conformation)) might result in higher affinity anti-HA FB.

Yes, we agree that it can be advantageous to have a higher affinity probe, but it is not always necessary. In fact, the original 15F11 scFv from which the frankenbody was derived had an affinity of 250 nM, yet was still useful for live-cell imaging applications. In general, we are very satisfied with the functionality of the anti-HA FB, particularly its ability to quickly light up single mRNA translation and 1x HA-tagged constructs. In our experience, extremely tight binding probes have more potential to interfere with the underlying biology. Thus, some turnover is often beneficial (Lyon and Stasevich, 2017). Turnover of probe can also minimize photobleaching to facilitate longer single molecule tracking (Viswanathan et al., 2015).

Due to the little bit lower affinity, the authors seemed to fuse 4x HA to POI to increase affinity by avidity effect. However, the long repeating HA tag might influence the conformation, folding, and/or localization of POI. Even, 10x HA encoding mRNA at the N-terminal might affect the translation efficiency and mRNA stability. The long HA tag could potentially exert appreciable perturbation, which can mislead the data. This reviewer wonders whether current FB-GFP can also efficiently colocalize 1x or 2x HA fused POI or mRNA. Further this reviewer wonder whether HA tag can be fused at the C-terminus of POI and 3'-UTR of mRNA. These results can validate the current results.

Thank you for pointing this out. The 4x tags were just carried over from our initial screen. We now demonstrate 1xHA fusion tags can also be imaged with HA frankenbody, including fusions to either the N- or C-terminus (Fig. 2a,c). We also show 1xHA-tagged proteins can be tracked at the single molecule level (Fig. 5 and S4) and 1xHA-tagged proteins can be visualized in zebrafish embryos (Fig. S7).

3. Fig. 2a: The authors stated that “frankenbodies can pass through the nuclear pore and bind target nuclear proteins”. FB-GFP does not have any nucleus localization signal. How FB-GFP can enter nucleus ? Based on the Fig. 2a data, FB-GFP binds to H2B in the cytosol, the complex of which then enters nucleus owing to H2B.

Since the FB is only ~50KDa, it can pass through the nuclear pore without an NLS (as shown by the even distribution of FB throughout cells lacking HA-tags, Fig. 1e). Thus, if a nuclear target exists, the FB spends most of its time bound to target in the nucleus. The idea that it is shuttled in with the H2B is also plausible, but may not be necessary.

4. Fig. 2d: The authors stated that “Encouragingly, all three frankenbody constructs colocalized with HA-tagged H2B in the nucleus of living U2OS cells, similar to the original GFP-tagged frankenbody (Fig. 2D, upper).” However, the results show that some portions of FB-Halo and FB-mCh remain in the cytosol.

The difference was due to the fact that FB-Halo and FB-mCh were over-expressed (CMV promoter) compared to the target HA-tagged protein (Ub promoter). This imbalance led to excess background FB with nothing to bind to. To correct for this, we replaced the Ub promoter on the HA-tagged protein with the CMV promoter. After doing this, all cells displayed little to no background signal, as shown and quantified in the updated Figs. 2e,f. These new data now verify our original statement “Encouragingly, all three frankenbody constructs colocalized with HA-tagged H2B in the nucleus of living U2OS cells, similar to the original GFP-tagged frankenbody”.

5. Fig. 2c: For a model of membrane protein (Kv2.1), the image does not clearly support the peripheral membrane localization. More sharp z-axis image is desirable or other transmembrane protein might be suitable as a model membrane protein.

To better demonstrate peripheral membrane localization, we constructed a 4xHA-mRuby-Kv2.1. With the improved construct, the colocalization of FB-GFP with HA-tagged Kv2.1 is now easy to see (Fig. 2b, right column). In addition, we also made a supplemental 3D projection movie to more clearly show FB-GFP labels HA-tagged Kv2.1 at the peripheral membrane (Movie S1).

Reviewer #2 (Remarks to the Author):

Report on manuscript entitled “A genetically encoded probe for imaging HA-tagged protein translation, localization, and dynamics in living cells and animal”, by Ning Zhao et al.

In this paper, the authors report the development of a monochain antibody against the HA tag (FB), which works in an intracellular environment. The antibody has an affinity in the 10 nM range in vitro, and an off rate of 2-3 minutes in vivo. It works not only as a

GFP fusion, but also when fused to mCherry, SNAP and HaloTag. When the protein of interest is fused to a repeated HA epitope, such as the spaghetti monster that contains 10xHA tag, the single chain antibody allows to visualize translation sites in living cell, as was previously reported for the SunTag. Finally, it is also suitable for labelling proteins in live zebrafish embryos.

Overall, the authors present a nice and useful tool to image translation and single molecule of proteins. There has been a strong need for good tools to label peptide tag in live cells, and this responds to this need. It is also highly complementary to the SunTag system that has been heavily used since its publication. The study is straightforward and well documented, and I strongly support publication in Nat Comm.

We thank the reviewer for the comments and feedback. Below we provide a point-by-point rebuttal. To facilitate the review, all of our responses and changes are shown with a dark-purple font.

The following comments should however be addressed:

1-Although the scFv is soluble with the tested proteins, the author should run a more demanding aggregation test, for instance using a 10xHA-mitochondrial protein, as done with the original SunTag paper. The absence of mitochondrial clustering would be a strong argument that the system does not lead to artifactual aggregation at high concentration.

To verify scFv solubility, we co-expressed FB-GFP with two new constructs: Mito-mCh-1xHA and Mito-mCh-smHA (10xHA) (note the mitochondrial protein is the same used in the SunTag paper). As now shown in Figs. 2c,d, FB-GFP colocalizes with Mito and does not lead to artifactual aggregation, even at the higher concentrations present with the 10xtag.

2-Figure 2c: the images of FB-GFP+smHA-Kv2.1 and GFP-Kv2.1 look quite different. Why not use a smHA-mCherry-Kv2.1 as in the other panels?

Thank you for the suggestion. We originally tried to make an mCh-Kv2.1 construct. However, for reasons which we still do not fully understand, mCh induces some mislocalization of Kv2.1. To correct for this, we have now made an improved construct with mRuby in place of mCh. With the improved construct, Fig. 2b now shows more definitively that HA-FB colocalizes with HA-tagged Kv2.1. We have also added a new supplemental 3D projection movie (Movie S1) of the representative neuron to more clearly show peripheral membrane localization.

3-Figure 2D: the efficiency of labelling seem to vary for the different constructs. Is this real or just an impression from the cells that were chosen in the images? if FB-GFP and FB-HaloTag are co-expressed, is the labelling similar for both fusions?

The difference was due to the fact that FB-Halo and FB-mCh were over-expressed (CMV promoter) compared to the target HA-tagged protein (Ub promoter). This imbalance led to excess background FB with nothing to bind to. To correct for this, we replaced the Ub promoter on the HA-tagged protein with the CMV promoter. After doing this, all cells displayed little to no background signal, as shown and quantified in the updated Figs. 2e,f. Furthermore, as the reviewer suggested, we have now also co-expressed FB-Halo, FB-SNAP, and FB-mCh with FB-GFP. In all cases, we get good labeling in both colors, suggesting the difference colored variants of FB bind with similar kinetics and can, in fact, compete with one another.

4-Figure 3b: I think that it would be a nice addition to the paper if the authors would use FB-HaloTag as a probe in fixed cells and demonstrate that this allows to perform STORM-type of super-resolution imaging.

We agree STORM-type imaging is possible with the FB. We now demonstrate this in living cells in Figs. 5 and S4. Specifically, we tracked tens of thousands of individual 1xHA-tagged histones to create a mobility map of histones across the cell nucleus. We believe this sort of application significantly strengthens the manuscript and further proves the diverse utility of the FB. We therefore thank the reviewer for the suggestion.

5-Figure 3C: the blot on the right is overexposed and should be replaced.

We have replaced the Western blot with an improved, non-saturated blot.

6-Figure 8. It is unclear from what is shown that the FB-GFP provides any advantage over the mCherry label fused to H2B. Can this experiment be repeated with for instance a 10xHA-sm? It would then perhaps highlight a strong gain in sensitivity of the FB approach.

We now demonstrate in Fig. S7 that that HA FB signal-to-noise improves with the number of repeat epitopes. With a 1x tag, the signal is worse than the mCh signal (left column); with a 4x tag, the two signals become comparable (middle column); finally, with a 10x spaghetti monster tag, HA FB signals are superior to mCh signals (right column).

7-In the discussion of motorized transport of polysome, the author could also cite ref21, which also report this phenomenon.

Thank you for pointing this out. We have now added the suggested reference (p.13, bottom paragraph and p.16, top paragraph).

REVIEWERS' COMMENTS:

Reviewer #1 (Remarks to the Author):

The authors properly addressed the raised concerns and comments by this reviewer.

Reviewer #2 (Remarks to the Author):

The authors have answered the comments and criticism of the reviewers, and they have added experiments further demonstrating the efficacy and versatility of their HA "frankenbody".

I believed that the authors developed a very nice tool for cellular imaging and I fully support publication of this manuscript.

REVIEWERS' COMMENTS:

Reviewer #1 (Remarks to the Author):

The authors properly addressed the raised concerns and comments by this reviewer.

Reviewer #2 (Remarks to the Author):

The authors have answered the comments and criticism of the reviewers, and they have added experiments further demonstrating the efficacy and versatility of their HA "frankenbody".

I believed that the authors developed a very nice tool for cellular imaging and I fully support publication of this manuscript.

We appreciate both reviewers' comments on our manuscript.